# Think Globally, Act Locally: Global Requirements and Local Transformation in Sugar Pots Manufacture in Sicily in the Medieval and Post-Medieval Periods

Roberta Mentesana [1,2,*], Anno Hein [3], Marisol Madrid i Fernàndez [1,2], Vassilis Kilikoglou [3] and Jaume Buxeda i Garrigós [1,2]

[1] Cultura Material i Arqueometria UB (ARQUB, GRACPE), Department d'Història i Arqueologia, Facultat de Geografia i Història, Universitat de Barcelona, 08001 Barcelona, Spain; mmadrid@ub.edu (M.M.i.F.); jbuxeda@ub.edu (J.B.i.G.)

[2] Institut d'Arqueologia de la Universitat de Barcelona (IAUB), Facultat de Geografia i Història, Universitat de Barcelona, 08001 Barcelona, Spain

[3] National Centre of Scientific Research "Demokritos", Institute of Nanoscience and Nanotechnology, 15310 Athens, Greece; a.hein@inn.demokritos.gr (A.H.); v.kilikoglou@inn.demokritos.gr (V.K.)

* Correspondence: r.mentesana@ub.edu

**Abstract:** Since medieval times, sugar production and consumption has had a huge impact on European social, cultural, and economic development. The introduction of sugar cultivation entailed knowledge transfer and new technological requirements, such as the manufacture of sugar pots used to crystallise sugar, which requires a specific design, and thermal and mechanical properties. This paper presents part of the results of the SPotEU project, funded under the Marie Skłodowska-Curie Actions, which explores the development and impact of sugar production in western Europe through the study of sugar pot manufacture from an interdisciplinary perspective, integrating archaeological and historical research with material science and material culture approaches. This paper focuses on sugar pots from Sicily, one of the main regions for sugar production in Western Europe in the 11–16th centuries A.D. Sugar pots were assessed from technological and performance points of view, aided by instrumental analysis (petrography, SEM, XRF, XRD, mechanical, and thermal property tests). The archaeological and analytical results are presented, revealing different centres of sugar pot production on the island, and specific choices in the design of the vessels and their properties. This allows us to discuss how craftspeople locally adapted their ceramic-making traditions to face the new product demands from the sugar production industry in the Mediterranean.

**Keywords:** ceramics; technological choices; petrography; SEM-EDX; WDXRF; PXRD; heat transfer properties; fracture strength

## 1. Introduction

In our industrialised and globalised world, separating the 'local' from the 'global' elements in the design and production of objects is a difficult matter. This is a difficulty that is not only practical but also emotional, as the concept of local is often paired with *tradition*, *identity*, *resilience*, and the *past*, while the global concept goes in the opposite direction [1]. Nevertheless, objects intended for the same function, produced industrially or not, tend to share strong similarities in design, although they probably will be highly varied if we can analyse their production sequence as we do with past material culture. If we can go beyond their common purpose and the impact this has on an object's appearance, we may discover that their production possibly involves raw materials, machinery, and product parts from different areas, and even different countries, and a blend of past and present design, but also different way of use. A can opener, now a common tool with the same functionality despite cultural or geographical contexts, embeds in its design and manufacture much more than what it may seem. If we compare a can opener sold in Italy and one sold in

the UK, they have commonalities, such as two rotating wheels (one cutting), two opposite levers, and a rotating handle. Probably both are ultimately made in Asia. The way these parts are combined creates two different objects, which imply a slightly different way of cutting: the first one cuts the can from the side (Figure 1a,c), whereas the other one from the top (Figure 1b,d). Neither of the two is technologically more advantageous (they are two of the multiple designs of the same object), and the first type was used in the UK until the 1980s, when the other type was invented and then rapidly adopted. One of the reasons for this may be that, in the UK, safety policy is included in everyday practices. This may have led to the acceptance of the top-cutting can opener, which allows opening a can without inserting the finger into the top part, risking cutting. Although familiar with the object *per se*, people living in the two countries may have difficulties in using a different type of can opener, because they would lack the body movement and mental structure required to operate it. Leaving aside the social acceptance factor, this example shows how similar objects, aimed at the same purpose, may have those small differences that tell us about product provenance, manufacturer design choices, and different mental attitudes and body gestures that required for operation. It tells us the ways in which a global requirement, i.e., opening a can, can be interpreted differently by manufacturers, even in our interconnected modernity, in response to different cultural attitudes.

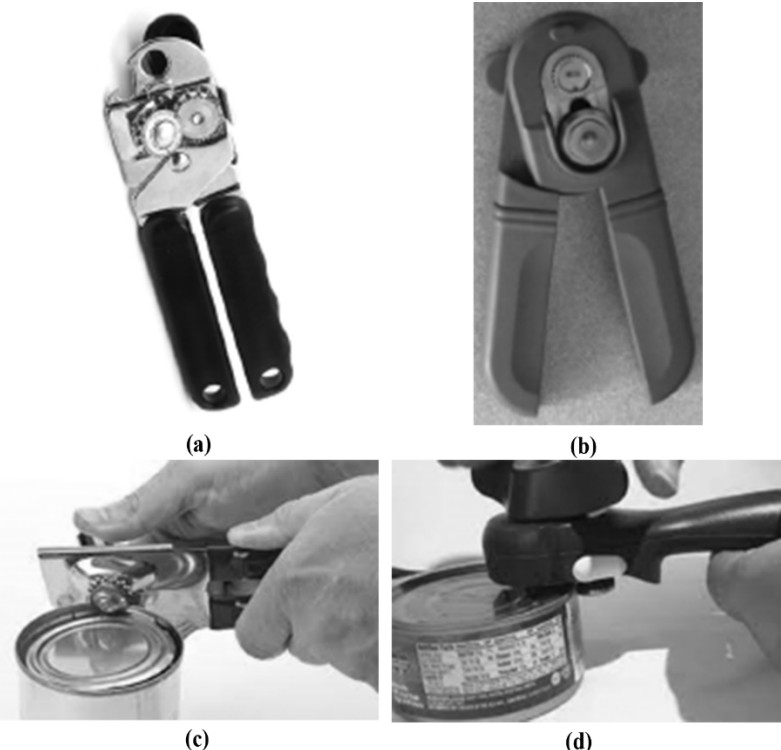

**Figure 1.** Design and use of a side cutting can opener (**a**,**c**) and a top cutting opener (**b**,**d**). Modified after (**a**) Evan-Amos, Public domain, via Wikimedia Commons, (**b**) © Materialscientist at English Wikipedia, CC BY-SA 3.0 https://creativecommons.org/licenses/by-sa/3.0 (accessed on 1 February 2022) via Wikimedia Commons, (**c**) © Whitestar1955 | Dreamstime.com https://koit.com/96-5-koit-blog/youve-using-can-opener-wrong-whole-time/ (accessed on 1 February 2022), (**d**) © Susan Brown https://www.youtube.com/watch?v=zePEyRB6Hqo (accessed on 1 February 2022).

The SPotEU project ("Sugar Pot manufacture in Western Europe in the medieval and post-medieval period (11–16th centuries AD)", funded under the Horizon 2020 Marie Skłodowska-Curie Actions (grant agreement: 797242)), of which this paper represents a part, was developed to explore the ways in which craftspeople faced the demands from the newly adopted sugar production in the Middle Ages. Although operating on a different scale than today, the Mediterranean has always been a medium for exchanging

materials, objects, and ideas, often transcending territorial political divisions and cultural differences [2,3]. Phenomena such as the production and consumption of sugar in the Middle Ages could be considered one of these pan-Mediterranean occurrences, which had, especially in the 14–16th century phases, a huge impact on European social, cultural, and economic development [4]. The chronology of the introduction of sugar cane into the Mediterranean suggests that it followed the Islamic expansion: it is documented in Egypt from the 8th century AD, and progressively in Cyprus, Crete, Sicily, the African coast, and the Iberian Peninsula by the 10th century AD [5]. However, what looks like a monolithic phenomenon linked with the movement of communities from the eastern Mediterranean may not have occurred in such a homogeneous way. Firstly, the 'Arab agricultural revolution', the label given to the set of intensive farming and irrigation technological novelties developed by Muslim groups [6], has lately been critically reviewed [7–9]. Moreover, it appears that in the Western Mediterranean, sugar production did not develop in the same way as in the Eastern Mediterranean. Regionally focused studies are now starting to reveal a more diversified picture of when and how the sugar industry was developed in each region, shedding light on local responses to the growing sugar demand [10–16]. Regarding commonalities, sugar production indeed required a specific sequence from its cultivation to processing, types of machinery (i.e., mill, a press, large firing installations), and objects having a specific purpose, such as cauldrons for sugar boiling and cone-shaped vessels for the crystallisation phase. Therefore, its introduction and cultivation entailed new technological requirements, and craftspeople and the local workforce needed to adapt or transform their skills and products to these new demands. Nevertheless, sugar production in the Western Mediterranean has often been observed from a top-down approach, from the point of view of merchant, landowner, royal, and ecclesiastical interests. Little is known about the role of craftspeople in the development of sugar production in modern Europe.

This paper explores the development and impact of sugar production in local craftsmanship, using one of the core crafts linked with sugar production as a baseline, i.e., the manufacture of sugar pots. These consist of a ceramic reversed cone with a hole at the bottom, where the liquid syrup is poured, and left to cool and crystallise, called the sugar mould cone (hereafter, *sugar cone*). They are often associated with molasses collecting jars (hereafter, *molasses jars*) where the liquid molasses, a discard of the first boiling of the sugar syrup, is collected when the sugar crystallised. The design and characteristics of this set of vessels (so-called *sugar pots*) developed explicitly for sugar production, and they are often the only archaeological evidence of past sugar production and consumption. Before the introduction of sugar pot manufacture, the design of these vessels was unknown to local potters, who faced several issues. First, the design of the sugar cone is dissimilar to that of other vessels: Ibn al-'Awwām, in his handbook on agriculture, refers to the specific shape for the sugar to crystallise [17] (p. 393), but he does not provide further specifications. Moreover, the dimension of the sugar cones seems to be linked to specific product quality: the smaller the size, the more refined the sugar they contained [18] (p. 276) [13] (pp. 59–77). In addition, sugar pots had to exhibit sufficient material properties and design to withstand repeated thermal changes and mechanical stresses; they were intended to be repeatedly used before discard and, therefore, we may suppose they were made to last. Potters had also to contend with the high volume of ceramics required, as vessels were frequently broken during the process and, in some other cases, shipped with the sugar content [13] (pp. 76–77). All these features may have been shared with those operating in sugar production, as suggested by the similarities in sugar pots across time and space [13] (cf. pp. 59–77). However, as in the case of the can opener, potters also may have developed different manufacturing strategies and organisations to meet these new demands.

Following a similar approach as that applied for sugar production materials in the Near East [13,15], this paper focuses on Sicilian medieval archaeological evidence for sugar production. For the first time, sugar pots from this region were examined in such technological detail that allowed us to assess the places of production, movement, and material properties of sugar pots. Specifically, this paper focuses on the micro-scale of the

potters' community, and their technological choices in manufacturing sugar pots within their cultural context of manufacture. To explore this, we first needed to identify local production, aided by chemical and petrographic characterisation. We then examined the material properties by means of microstructure and textural analysis, and mineralogy, and how the materials affect the thermal and mechanical properties of the vessels [19–21]. This allowed us to discuss whether potters produced vessels with different characteristics. Considering their context of use, sugar pots require resistance to the thermal stresses of the boiling sugar syrup poured in to them (thermal shock resistance), but also must allow the dissipation of heat at a certain rate to allow the sugar to crystalise (heat transfer). In addition, sugar cones can be stacked directly on molasses jars or placed over a hole in a wooden bench, and were also frequently handled for production; in some cases, these pots travelled long distances, either empty or filled with sugar. Sugar pots, and specifically sugar cones, would need to withstand forces applied to them without failing (fracture strength and toughness). The mechanical and thermal properties were used as predictors of the performance of these vessels during use [22] because many parameters influence each other, and these parameters cannot be disentangled in archaeological ceramics, as can be done during experiments [23]. A further step, which will be published in another paper, simulated whether these vessels perform in the same way under similar use conditions [24,25]. This complex and multiphase set of examinations did not aim to assess functionality; rather, the objective was to explore the reasons for these choices, other than performance [23]. As well explained by Sillar and Tite [26] (p. 4), 'it is impossible to account for any of these choices without combining a consideration of both the material properties and the cultural context'.

## 2. Materials

According to the archive sources, authors have distinguished different phases in the sugar production on the island [18,27–29]. From the first acknowledgement in the mid-10th century until the beginning of the 13th century A.D., the production seemed concentrated in Palermo, and was small scale and mainly devoted to the consumption of the higher class and the pharmacopoeia. A crisis may have occurred before the 13th century if Frederick II had tried to reinstall Palermo's production. At the beginning of the 14th century, and particularly during this century, sugar production intensified. The number of sugar production centres increased considerably, leaving Palermo's walls and extending to the entire island. Sugar cane cultivation started taking up extensive portions of land, although co-existing with other crops. It was transformed into on-purpose large buildings (*trapetum,-*), involving several specialised workers, and exported to the whole European market. An economic crisis may have occurred between the end of the 15th century and the beginning of the 16th century, but sugar production flourished again in the mid-16th century until its final collapse in the mid-17th century. From this summary, it is clear that sugar production in Sicily had a long but discontinuous development. It could be wondered whether, during these phases, knowledge about sugar production and the manufacture of sugar pots was transmitted, and in which manner.

The archaeological evidence of sugar production on the island is concentrated in the north-western part of Sicily (Figure 2). A recent diachronic examination of the archaeological evidence [16] suggests that, despite the chronological issues, the design and volume of sugar pots changed with the phases noted above. However, they are also site-specific; that is, even sites of the same phase show differences in the design of the pots. Materials from some of these sites were studied and will be discussed extensively in a forthcoming publication [16], and are briefly presented here.

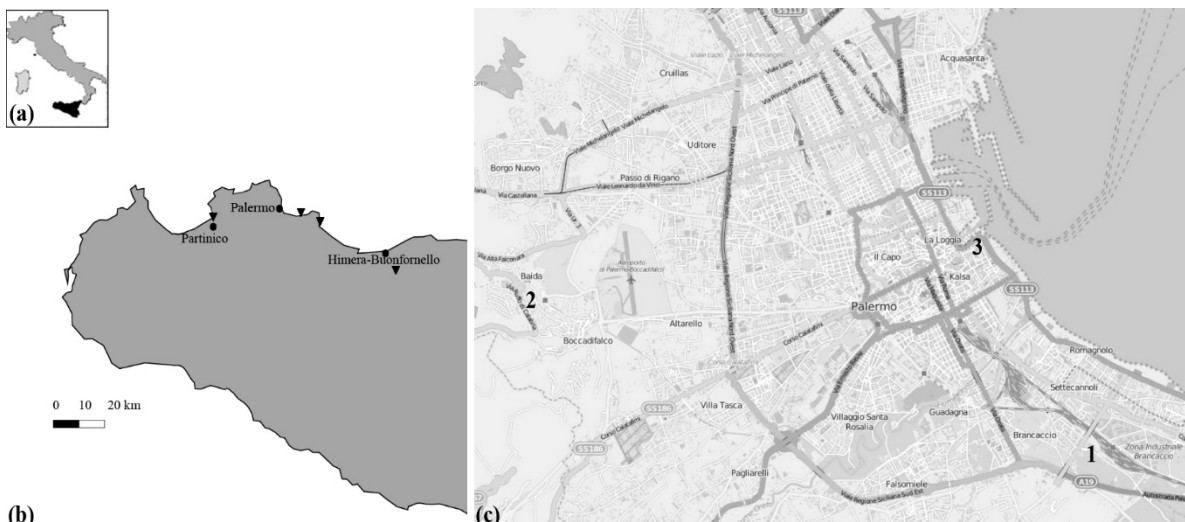

**Figure 2.** Map of Italy (**a**); western Sicily with the site discussed (**b**) (black dots: site sampled, black triangles: geological deposits sampled); and map of Palermo (**c**), with the sites discussed in the text. 1: Castello della Favara in Maredolce, 2: Convento di Baida, 3: Palazzo Steri–Chiaramonte.

(1) The Castello della Favara in Maredolce (MAR), in Palermo, was built with the artificial lake by the Norman king Roger II; in the 14–15th centuries, the building changed its function from residential to agricultural/industrial, and was also linked with sugar cane production [30] (p. 473) [28] (p. 113). Here sugar pots were found during the excavation of a filling of the lake of Maredolce [30,31]. They were dated to an earlier phase of sugar production and consumption, before the end of the 13th century [16].

(2) The excavation of Palazzo Steri–Chiaramonte (STE), in the current Piazza Marina in Palermo, by Tusa [32], revealed a long sequence of occupation of the site, which was the residence of many Sicilian rulers from the 14th century. In layers corresponding to the phase from the end of the 15th century to the beginning of the 16th century [33,34], some sugar pots were found, probably related to sugar consumption in the palace, which at that time was the residence of the Viceré.

(3) A large sugar cone was also found in the storeroom of the archaeological museum of Palermo "A. Salinas", and is related to an underwater finding (UND) in Palermo's waters.

(4) Another seven cones were found in the same museum; these were retrieved during the restoration of the Convento di San Giovanni of Baida (BAD), where the cones were used as the filling material of the vaults. The latter may tentatively be dated to before the end of the 15th century [16].

(5) West of Palermo, the area of Partinico (PAR) was intensively dedicated to wine and sugar cane plantations since the end of the 14th century [35] (pp. 41–42) [27] (pp. 101–111). Here, some sugar pots were recovered during a survey and, therefore, only dated to after the end of the 14th century, which is the first known date of archive sources for sugar production in the area.

(6) Lastly, during the excavation of the Greek colony of Himera (HIM), near the present-day town of Termini Imerese, conspicuous traces of a sugar production site were found, with numerous sherds of sugar pots only partly recovered [36,37]. The few glazed ceramic sherds allow us to date these vessels to between the end of the 15th century and the beginning of the 16th century [16,38].

In summary, the sugar pots examined in this paper belong to different chronological phases: one between the 11th and 13th centuries, to which only Castello della Favara a Maredolce could be placed; and another from the end of the 14th to the beginning of the 16th centuries, where all the other contexts belong, most of them to the range from the end of the 15th century to the beginning of the 16th century.

Sugar pots from these sites were measured and grouped into types based on their rim features and the typology previously used by Falsone for molasses jars [32] (Figure 3). A total of 87 sugar pots (26 molasses jars and 62 sugar cones), in addition to five vessels belonging to other types for comparison, were sampled for analytical study as representative of each site, chronological phase, and type (Table 1) (These comparison types were noria vessels and cantaro. Noria vessels are usually bell-shaped vessels, with a pointy bottom, used as water buckets on a waterwheel (noria), whereas a cantaro is a two-handled cylindrical vessel. Both these shapes were found in conspicuous quantities at the Palazzo Steri–Chiaramonte site [33,34]). Individuals were labelled with an acronym indicating the location where they were found and a progressive number (Table S1). The complete list of the individuals studied is published and openly accessible [39].

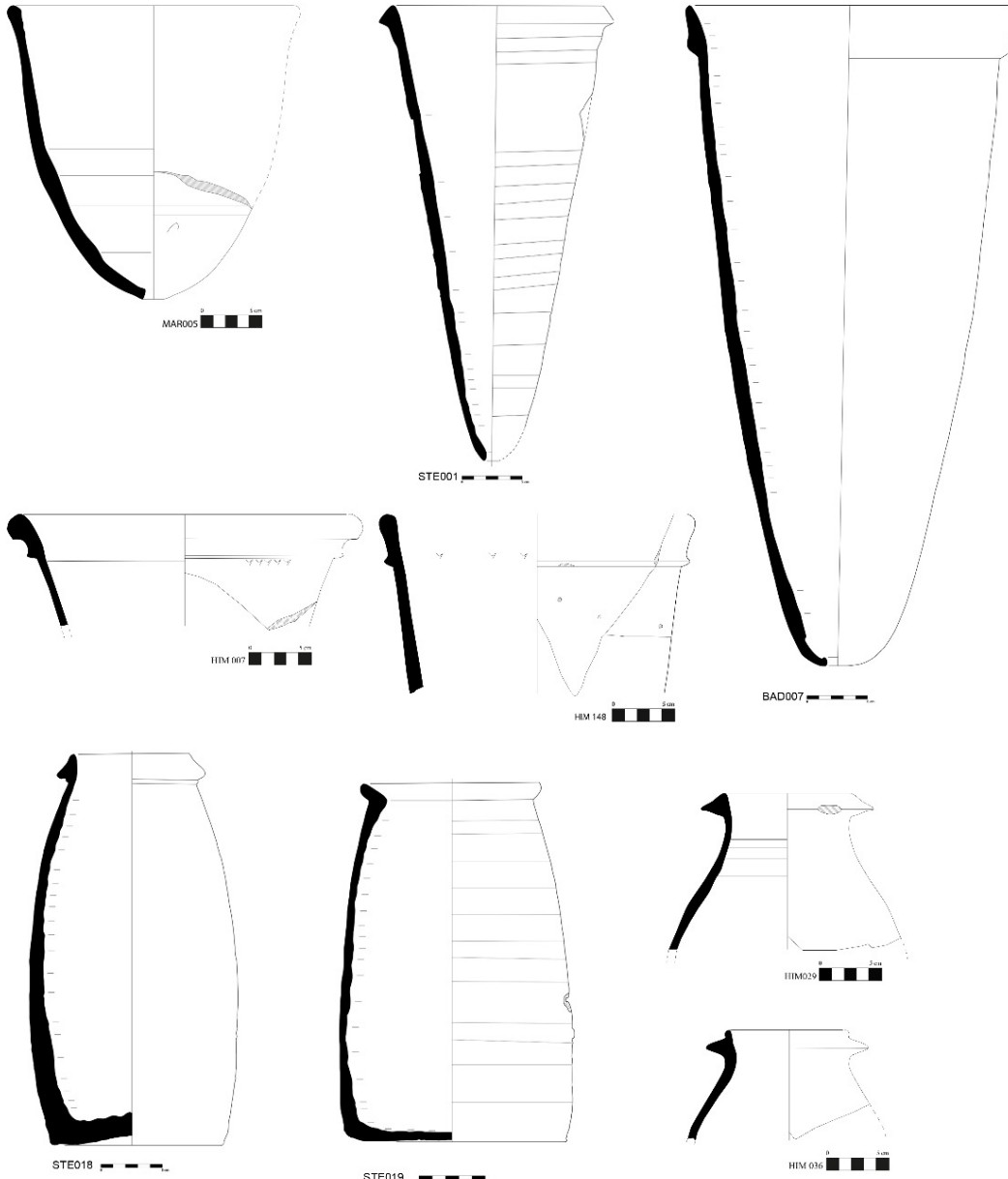

**Figure 3.** Representative sugar cones and molasses jars for each type: sugar cone type 1 (MAR005), type 2 (STE001), type 3 (BAD007), type 4 (HIM007), type 5 (HIM148); molasses jar type A.1 (STE018), type A.2 (HIM029), type A.3 (HIM036), type B (STE019). (Drawings of STE001, BAD007, STE018 and STE019 by S. Arrabito, others by R. Mentesana).

**Table 1.** Archaeological sites studied and sampled for this study.

| Site | Vessel Shape | | | Tot. Site |
|---|---|---|---|---|
| | Molasses Jar | Sugar Cone | Others | |
| Underwater, Palermo (UND) | | 1 | | 1 |
| sampled | | 1 | | 1 |
| Castello della Favara in Maredolce, Palermo (MAR) | 2 | 8 | | 10 |
| sampled | 2 | 8 | | 10 |
| Convento di Baida, Palermo (BAD) | | 7 | | 7 |
| sampled | | 5 | | 5 |
| Palazzo Steri–Chiaramonte, Palermo (STE) | 5 | 7 | 7 | 19 |
| sampled | 2 | 7 | 5 | 14 |
| San Giovanni degli Eremiti? (PAM) | | 1 | | 1 |
| sampled | | 0 | | 0 |
| Himera–Buonfornello (HIM) | 50 | 126 | | 176 |
| sampled | 19 | 34 | | 53 |
| Partinico (PAR) | 3 | 8 | | 11 |
| sampled | 3 | 7 | | 10 |
| Tot. studied | 60 | 156 | 7 | 225 |
| Tot. sampled | 26 | 62 | 5 | 93 |

Geological deposits near the sites under study were sampled to compare them with the raw materials of the archaeological individuals, and to make the experimental briquettes needed for the mechanical and thermal tests (Table 2). DHIM01 was collected at the present-day tile industry of Later Siciliana (Collesano, Termini Imerese), where deposits of Terravecchia formation clay can be found. Two deposits of Ficarazzi formation were sampled: DPAM01 was collected on the beach of Ficarazzi, Palermo; DPAM02 was gathered on the beach of Santa Flavia (Palermo) near the archaeological site of Solunto. Lastly, for Partinico, some deposits were sampled at the Baia di San Cataldo; only one resulted in being adapted for ceramic making, and corresponds to a Numidian flysch formation. Except for the deposit from Partinico, those of Ficarazzi and Terravecchia formations have been extensively characterised [40,41].

**Table 2.** Geological deposits sampled for this study.

| ID | Location | Geological Formation |
|---|---|---|
| DHIM01 | Collesano (PA) 37°57′48.8″ N 13°50′58.8″ E | Terravecchia |
| DPAM01 | Ficarazzi (PA), 38°05′44.1″ N 13°27′23.3″ E | Ficarazzi |
| DPAM02 | Santa Flavia (PA) 38°04′53.1″ N 13°32′14.1″ E | Ficarazzi |
| DPAR01 | Baia di San Cataldo, cala dei Muletti (PA), 38°05′06.0″ N 13°05′02.0″ E | Numidian flysch |

## 3. Methods

### 3.1. Theoretical Framework

Sugar pots started to be produced with the introduction of sugar cultivation in Sicily. The design and the property requirements of these vessels were unknown to potters before this time. Sugar pots were made with the same intended function as in other places in the Mediterranean where sugar was produced; it may thus be wondered whether this implies a similarity in the material requirements, and therefore, the choices of potters. The link between material properties, shape, and function has been advocated in many

studies: choices were directed towards creating the best-fitting, best-performant solution [42–46]. It has been argued that material culture is adapted to different natural or cultural circumstances, and survives and is transmitted according to its *fittingness* [47]. The evolutionary explanation has been successfully adopted to explain changes in the ceramic making in different contexts [48–50]. Many studies interpret their findings in this way, even if it is not clearly stated. Some of the authors share some aspects with the systemic approach [51], whereas others consider that this approach does not encompass other reasons that come into play during manufacturing processes. Technological choices, as formulated by Lemonnier [52], represent the conscious and unconscious adoption of certain technical features, and the dismissal others, based on multiple intermingled factors, such as material properties, ways of doing, the environment, beliefs, and, in short, the entire socio-cultural system of the individual and the group. This concept goes past the division between technique and technology, and the functional and style aspects of material culture, thus allowing an understanding of the material culture as a whole. However, it becomes difficult to translate the technological choice approach into a methodology for examining these choices, especially if it implies analytical techniques. Jones [53], amongst other authors [54,55], discusses the challenges of interpreting human actions starting from analytical data, but he argues that the finer the analytical technique used (i.e., chemical characterisation), the greater the tendency to interpret the results within the framework of general histories (macroscale); and the coarser the technique (i.e., macro-observations), the more the results will be interpreted as a local development (microscale). It has been argued elsewhere [56] that, rather than the analytical techniques, the research objectives define the scale of analyses—macro or micro—and that *chaîne opératoire* [57] can be one of the best operational frameworks to overcome the issue of merging different data types and scales of research. The *chaîne opératoire* approach helps in following a structured reconstruction of the sequence of ceramic manufacture, which goes from raw material selection and manipulation, to forming, surface treatment, and firing, and is able to incorporate elemental to macroscopic data [58]. The *chaîne opératoire* approach is a useful framework to understand the manufacturing sequence, but becomes meaningful only when embedded in the learning context of *practice*. This concept, created by Bourdieu [59,60] and transformed by Lave and Wegner into the *community of practice* [61,62], allows us to understand the way in which technological choices are generated and reproduced by people in their everyday practical activities. In our case study, dealing with an input external to ceramic manufacture, the communities of practice approach can help understand how individuals and collective choices are created and negotiated through the already existing ceramic manufacturing practices.

### 3.2. Analytical Approach

Petrographic examination (PE), wavelength dispersive X-ray fluorescence (WD-XRF), and powder X-ray diffraction (PXRD) analyses were used to characterise the petrographic, chemical, and mineralogical composition of the paste. PE was also used to infer forming techniques coupled with a macroscopic examination of forming traces [63]. Scanning electron microscopy with energy-dispersive X-ray spectroscopy (SEM-EDX) enabled the study of the microstructure, estimation of the vitrification stage, and microanalysis of features of interest [64,65]. The combination of these different techniques allowed the study of provenance, raw material manipulation, forming, firing regimes, and surface treatments. All the ceramic samples and the fired geological deposits were analysed by WDXRF, PE, and PXRD. In addition, in a multiphase sampling strategy, a subsample of 20 previously analysed individuals was sampled for study under SEM-EDX, according to the classification revealed by the XRF, PE, and XRD analyses in terms of meaningful compositional groups and mineralogical fabrics [51]. Both archaeological and experimental ceramics were characterised and tested for mechanical and thermal properties to investigate textural and microstructural variables and their impact on material performance. Sample

preparation and instrumental conditions are available in the Supplementary Material 2 [20–22,35,40,41,65–77].

## 4. Results

*4.1. Macroscopic Examination, Chemical, Mineral-Petrological, and Microstructural Results*

The results of elemental concentrations of the individuals analysed by WD-XRF [78] correspond with a special case of the projective $d$ + 1-dimensional space where the projective points are projected into the simplex $\mathbb{S}^d$. Points are represented by homogeneous coordinates that have a constant sum $k$ ($k \in R_+$):

$$C\,(\mathbf{w}) = \mathbf{x} = [x_1, \dots, x_d, x_{d+1}] \mid x_i \geq 0\ (i = 1, \dots, d, d+1),\ x_1 + \dots + x_d + x_{d+1} = k,$$

(in this case, $k = 100$). The projective points' vector space is the positive orthant. Hence, for the statistical data treatment, the raw concentrations were additive log-ratio (alr) transformed, according to:

$$\mathbf{x} \in \mathbb{S}^d \to \mathbf{y} = \ln\left(\frac{\mathbf{x}_d}{x_{d+1}}\right) \in \mathbb{R}^d$$

where $\mathbb{S}^d$ is the d-dimensional simplex and $\mathbf{x}_d = [x_1, \dots, x_d]$. They were also centred log-ratio (clr) transformed following the equation:

$$\mathbf{x} \in \mathbb{S}^d \to \mathbf{z} = \ln\left(\frac{\mathbf{x}}{g(\mathbf{x})}\right) \in \mathbb{H} \subset \mathbb{R}^{d+1}$$

where $\mathbb{S}^d$ is the d-dimensional simplex, $g(\mathbf{x})$ the geometric mean of all $d$ + 1 components of $\mathbf{x}$, and $\mathbb{H} \subset \mathbb{R}^{d+1}$ a hyperplane vector subspace of $\mathbb{R}^{d+1}$ [67,79–81].

The statistical treatment was performed on 88 archaeological individuals (HIM015, 029, 070, 072, 146 were removed as only major and minor elements were measured) and the four geological deposits, for a total of 92 individuals. The statistical data treatment of the chemical data was performed on the retained values using R [82], and the first step was to measure the existing variability in the dataset. This variability results from the difference in the chemical data, and how evenly the chemical differences relate to the retained components [51,83]. In this case, total variation (tv = 0.75) was higher than expected for a monogenetic set [84]. Focusing our attention on the compositional evenness graph (Figure 4), most of the variability is linked to the relative concentrations of CaO, MnO (tv/$\tau_{.j}$ < 0.3), and to a lesser extent, to $Na_2O$, $K_2O$, Rb, Ga, and Zr (0.3 < tv/$\tau_{.j}$ < 0.5). The compositional evenness was measured according to information entropy ($H_2$), also known as the Shannon index [85], on the $\tau_{.j}$ values in decreasing order [51]. Looking at the scatterplot matrix using the three clr-transformed components that introduce more variability, CaO, MnO, and $Na_2O$ (Figure 5), the distribution of samples into two groups seems clear; one of these corresponds to the samples recovered from Himera–Buonfornello, and the other from the different sites in Palermo. In addition, the plot $\ln(CaO/g(\mathbf{x}))$ vs. $\ln(MnO/g(\mathbf{x}))$ also shows a few samples placed together that may form a small group related to Partinico.

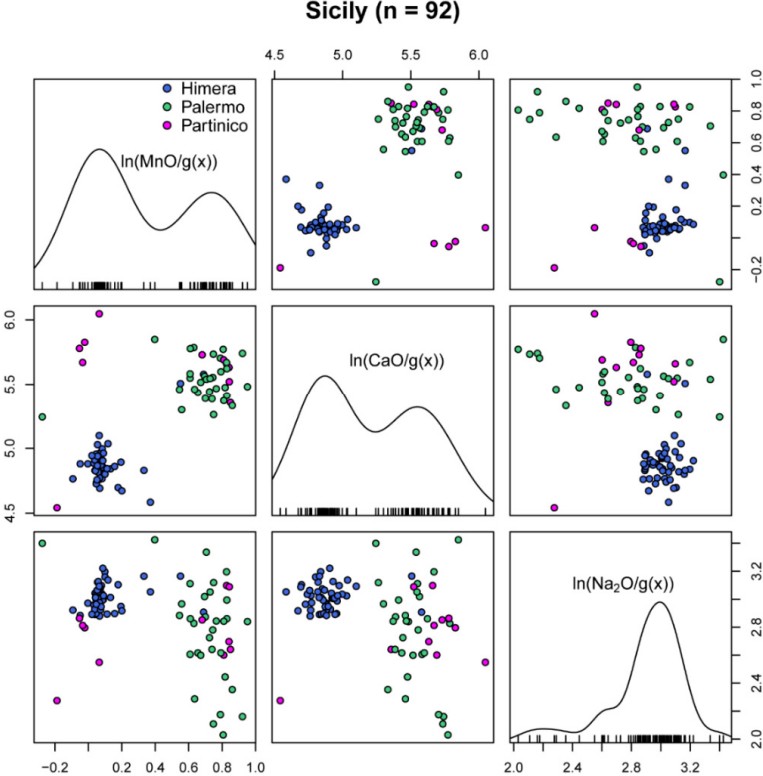

**Figure 4.** Compositional evenness graph of all the measured individuals according to the entropy value ($H_2$ and its relative value from the maximum attainable, $H_2$ %) of the elements considered for the statistical treatment of data. tv = total variation. $\tau_{.j}$: trace of the variance-covariance matrix following the alr transformation using that *j* element as the divisor. Vertical dotted lines express different tv/$\tau_{.j}$ values.

**Figure 5.** Scatterplot matrix on clr-transformed data using the three components that introduce most variability: MnO, CaO, and Na$_2$O. In the diagonal, KDE of these clr-transformed components. Individuals are grouped by site found.

As a second step of exploring chemical data, we provide the dendrogram from the cluster analyses (Figure 6) and the form and covariance biplots (Figure 7). The dendrogram presents results from the clr-transformed data, using the square Euclidean distance and the centroid agglomerative algorithm; whereas the form and covariance biplots present results from the singular value decomposition of the double-centred clr transformation [86–88]. The resulting form and covariance biplots of the first two principal components explain more than 70% of the variance (VE = 72.84%). Both in the dendrogram and the biplots, a structure of four groups and five ungrouped individuals can be observed: CGHIM01, CGPAR01, CGPAL01, and CGPAL02. The same structure of groups is observed when treating the data independently by finding location, that is, individuals recovered in Palermo and Partinico together separated from those from Himera–Buonfornello. According to the biplots, the most significant components in this discrimination are those already revealed by the compositional evenness graph of Figure 4. The first component exhibits the opposition of CaO, MnO, and Zr in the negative values, and $K_2O$, Rb, and Ga in the positive values, which is responsible for the distinction of CGHIM01 from the other groups. For the second component, mainly $Na_2O$, Zn, and Sr in the positive values are responsible for the discrimination of groups CGPAL01 from CGPAL02, and for the intra-group differences of the group CGPAL02. Regarding the ungrouped individuals, those corresponding to the clay samples DPAR01, DPAL01-02 remain isolated, whereas the one from underwater recovery is plotted close to CGPAR01 and CGPAL02, indicating compositional similarities (Table 3).

**Table 3.** Mean ($\bar{x}$), standard deviation (s), and total variation (tv) of the groups of more than two samples and values of the loners (as normalised values). Major and minor elements (expressed as oxides) in $w$%. Trace elements in μg/g.

| | DPAR01 | DPAM02 | DPAM01 | UND001 | CGPAR01 (*n* = 4) | | CGPAL01 (*n* = 6) | | CGPAL02 (*n* = 31) | | CGHIM01 (*n* = 47) | |
|---|---|---|---|---|---|---|---|---|---|---|---|---|
| | | | | | $\bar{x}$ | s | $\bar{x}$ | s | $\bar{x}$ | s | $\bar{x}$ | s |
| $Na_2O$ | 0.48 | 1.6 | 1.25 | 1.19 | 0.73 | 0.13 | 0.46 | 0.08 | 0.88 | 0.2 | 1.16 | 0.11 |
| MgO | 1.99 | 2.14 | 1.24 | 6.28 | 2.18 | 0.06 | 2.33 | 0.19 | 1.87 | 0.28 | 2.66 | 0.17 |
| $Al_2O_3$ | 17.97 | 17.33 | 9.59 | 14.93 | 12.76 | 0.54 | 14.35 | 0.93 | 13.96 | 0.85 | 17.77 | 0.62 |
| $SiO_2$ | 64.04 | 58.42 | 66.42 | 57.8 | 60.00 | 1.08 | 59.24 | 1.18 | 61.51 | 1.44 | 59.91 | 1.28 |
| $K_2O$ | 2.24 | 2.29 | 1.42 | 1.65 | 1.78 | 0.16 | 1.64 | 0.11 | 1.73 | 0.31 | 2.92 | 0.30 |
| CaO | 4.61 | 10.1 | 14.13 | 10.17 | 15.79 | 1.83 | 14.18 | 1.57 | 12.62 | 1.5 | 7.37 | 0.79 |
| $TiO_2$ | 0.97 | 0.93 | 0.64 | 0.85 | 0.72 | 0.03 | 0.81 | 0.04 | 0.79 | 0.05 | 0.90 | 0.03 |
| V | 147 | 143 | 87 | 100 | 101 | 7 | 96 | 5 | 86 | 6 | 134 | 9 |
| Cr | 125 | 121 | 88 | 89 | 81 | 11 | 87 | 4 | 83 | 8 | 125 | 7 |
| MnO | 0.04 | 0.04 | 0.06 | 0.11 | 0.05 | 0.00 | 0.11 | 0.01 | 0.10 | 0.01 | 0.06 | 0.01 |
| $Fe_2O_3$ | 7.52 | 7.00 | 5.13 | 6.88 | 5.85 | 0.29 | 6.74 | 0.38 | 6.39 | 0.37 | 7.08 | 0.21 |
| Ni | 39 | 35 | 20 | 42 | 30 | 2 | 40 | 2 | 37 | 4 | 49 | 3 |
| Zn | 91 | 88 | 52 | 78 | 70 | 6 | 78 | 3 | 96 | 27 | 104 | 4 |
| Ga | 21 | 19 | 9 | 14 | 13 | 1 | 14 | 1 | 13 | 2 | 22 | 1 |
| Rb | 108 | 101 | 62 | 62 | 68 | 5 | 73 | 2 | 71 | 10 | 124 | 7 |
| Sr | 181 | 344 | 348 | 290 | 340 | 18 | 312 | 57 | 274 | 37 | 322 | 29 |
| Y | 27 | 26 | 20 | 27 | 23 | 1 | 30 | 2 | 30 | 2 | 27 | 1 |
| Zr | 251 | 225 | 287 | 248 | 232 | 12 | 265 | 14 | 276 | 13 | 212 | 10 |
| Nb | 29 | 26 | 18 | 23 | 22 | 1 | 24 | 2 | 23 | 1 | 25 | 1 |
| Ba | 304 | 224 | 216 | 284 | 370 | 55 | 326 | 41 | 345 | 52 | 363 | 44 |
| Ce | 105 | 104 | 49 | 80 | 72 | 12 | 87 | 9 | 85 | 7 | 90 | 6 |
| Th | 15 | 15 | 11 | 14 | 14 | 1 | 16 | 1 | 14 | 1 | 16 | 1 |
| tv | | | | | 0.12 | | 0.13 | | 0.33 | | 0.08 | |

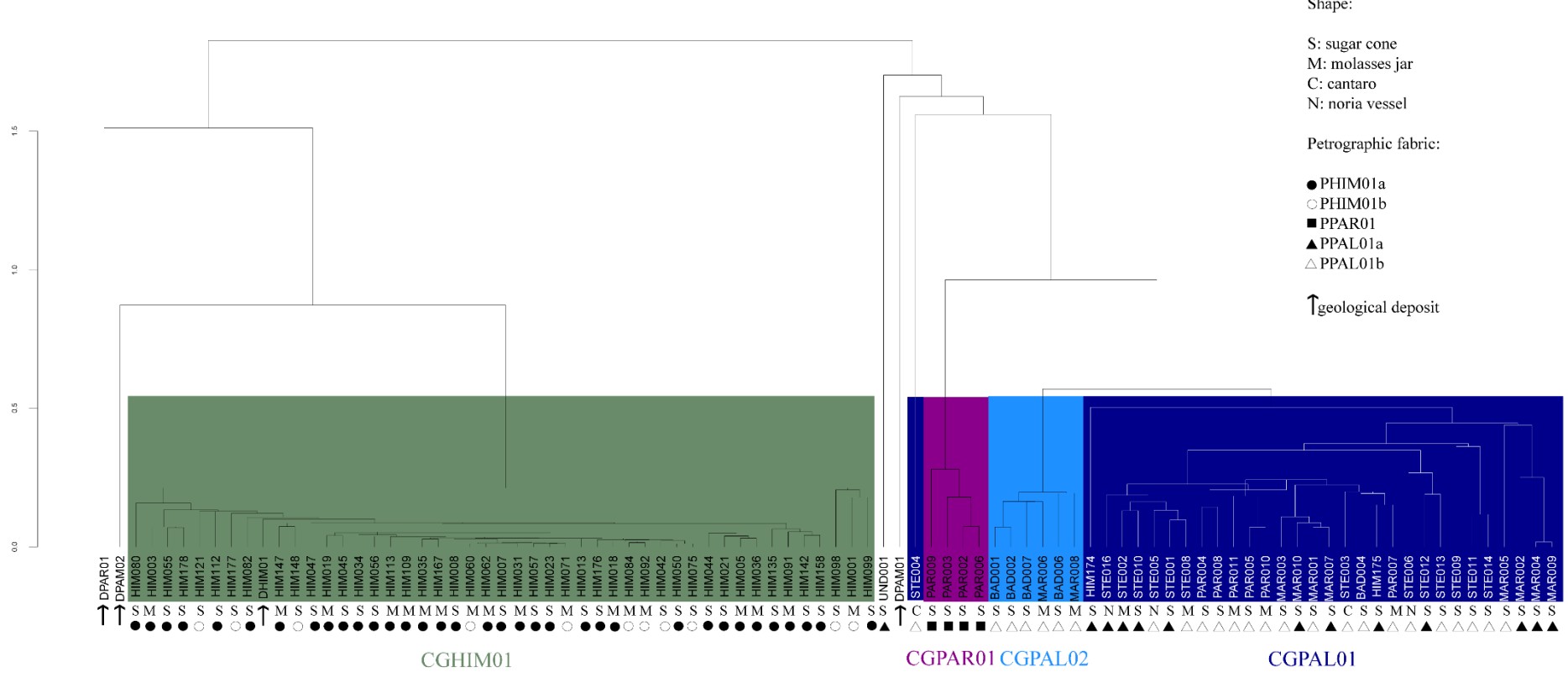

**Figure 6.** Dendrogram resulting from the analysis of the 92 individuals on the sub-composition Na$_2$O, MgO, Al$_2$O$_3$, SiO$_2$, K$_2$O, CaO, TiO$_2$, V, Cr, MnO, Fe$_2$O$_3$, Ni, Zn, Ga, Rb, Sr, Y, Zr, Nb, Ba Ce, and Th, clr transformed, using the square Euclidean distance and the centroid agglomerative algorithm.

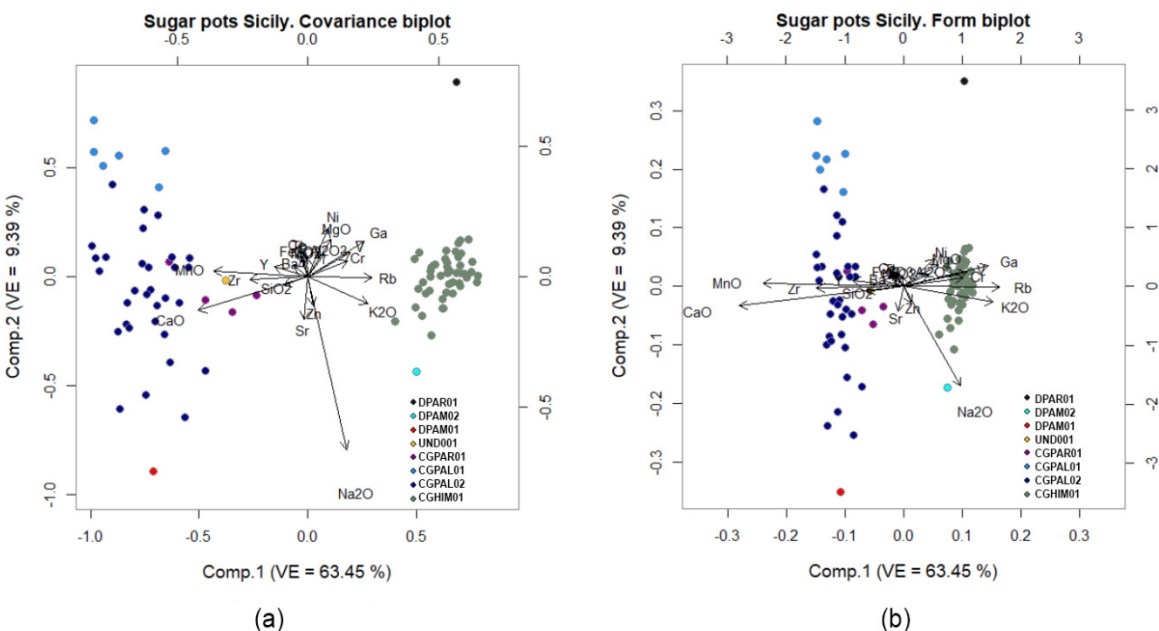

**Figure 7.** Covariance (**a**) and form (**b**) biplots of the principal component analysis using the singular value decomposition of the double-centred clr-transformed sub-composition $Na_2O$, MgO, $Al_2O_3$, $SiO_2$, $K_2O$, CaO, $TiO_2$, V, Cr, MnO, $Fe_2O_3$, Ni, Zn, Ga, Rb, Sr, Y, Zr, Nb, Ba, Ce, and Th.

The first large group on the left of the dendrogram is composed of 47 individuals recovered from the production centre of Himera–Buonfornello, except one individual that corresponds to a geological deposit (DHIM01), collected near the site. This group is very homogeneous and can be considered calcareous, although the CaO content is not very high (around 7 *w*%, Table 3). PE confirmed this grouping, as all these individuals were grouped in fabric PHIM01 (Supplementary Material 2). This is a fine- to medium-grained fabric composed mainly of monocrystalline quartz, feldspars, and mica in both the fine and coarse fraction; some sedimentary rocks (quartzarkose to quartzwacke, limestone, and chert) and metamorphic rock fragments (quartzite) are frequently to scarcely present in the coarser fraction. This fabric was divided into two sub-fabrics on the basis of the size distribution and frequency of the coarse fraction, larger and more homogeneously distributed in PHIM01b (Figure 8a,b). These differences may reasonably be due to intra-source variation rather than to technological reasons, as the sample shows a continuum of this feature, sometimes without a clear division amongst the two subgroups. This subdivision is not mirrored in the dendrogram resulting from the statistical manipulation of chemical data (Figure 6). The five individuals from the same site for which only major and minor elements were measured fit into this group both chemically (related only to these elements) and petrographically. The fabric is highly compatible with Terravecchia formation deposits available near the site [41] and the collected one, DHIM01, is strongly comparable with the ceramic samples, especially with subgroup PHIM01a (Figure 8g), as chemical data previously showed. Both sugar cone and molasses jars are included in this group (Tables S2 and S3).

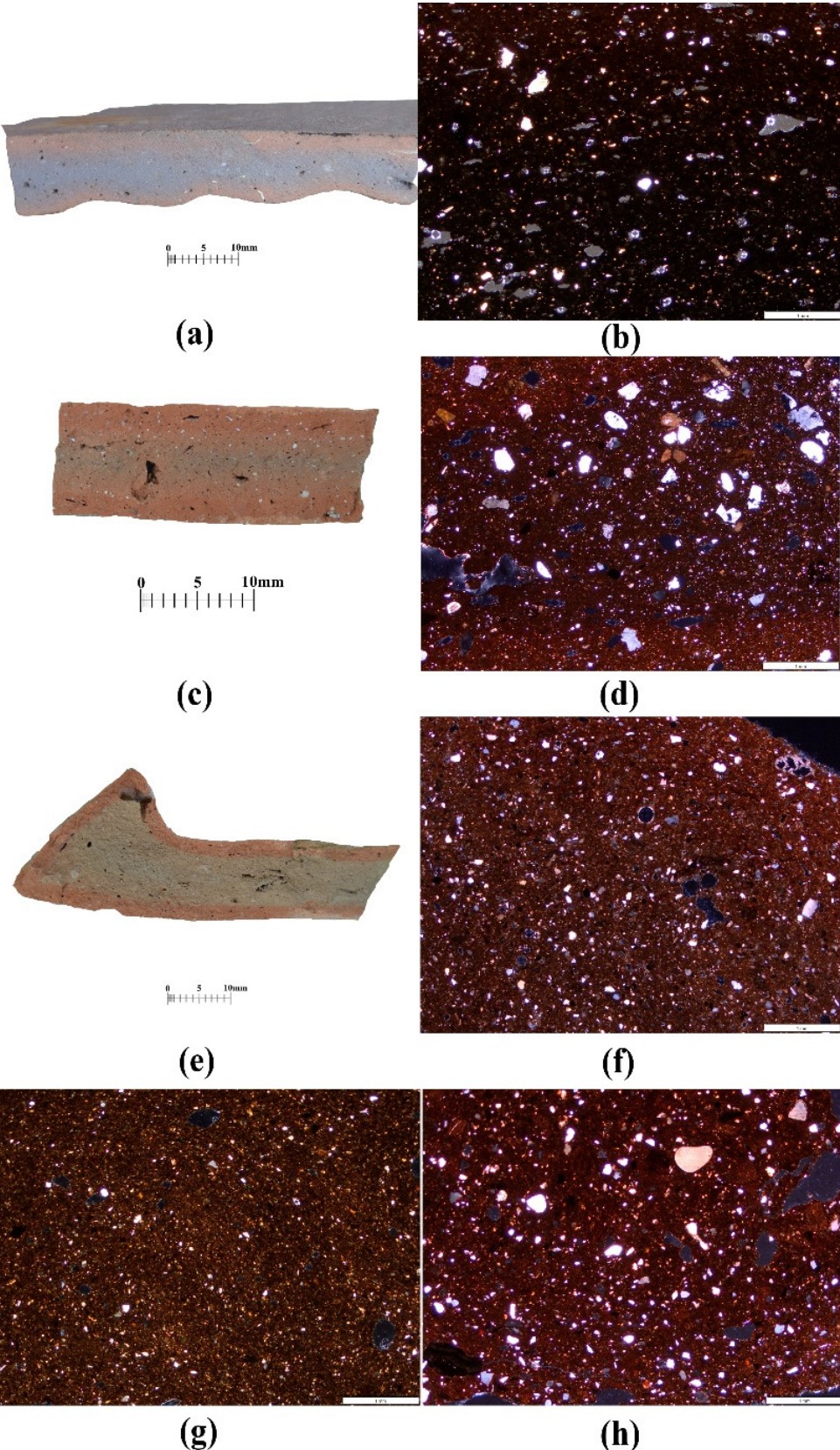

**Figure 8.** Fabric PHIM01a: macrophotograph (**a**) and photomicrograph in XP (**b**) of individual HIM178; fabric PHIM01b: macrophotograph (**c**) and photomicrograph in XP (**d**) of individual HIM078; fabric PPAR01: macrophotograph (**e**) and photomicrograph in XP (**f**) of individual PAR002; photomicrograph of DHIM01 (**g**) and DPAR01 (**h**) fired at 850 °C.

A group of four of the nine individuals from Partinico included in this study join together in the centre of the dendrogram (CGPAR01), showing clear differences from Palermo individuals, placed to the right (Figure 6). Covariance and form biplots confirm the discrimination of this small group, although its similarities with the individuals from Palermo, especially in the CaO content, placed both groups very close to each other. These individuals were grouped in the fabric PPAR01, which is a fine-grained fabric, mainly characterised by quartz and feldspars in both the fine and coarse fractions with a coarser fraction, when present, characterised also by chert, quartzite, and limestone; microfossils are common (Figure 8e,f; Supplementary Material 2). The fabric may be compatible with some deposits present in the area of Partinico [75]. However, the Numidian flysch geological sample, DPAR01, did not provide a clear matching fabric for the archaeological individuals in terms of composition and grain size distribution (Figure 8h). In addition, chemical results show that this deposit differs in multiple elemental concentrations from individuals in CGPAR01, amongst which it strikes the lower concentration in CaO and Sr, and the highest in Cr and Ce, compared to the archaeological individuals (Table 3). The composition of this deposit matches other Numidian flysch deposits analysed from western Sicily [40,41] which differs from the archaeological individuals grouped in this study. Montana et al. [41] (p. 96) note that the clay deposits collected in Palermo show a higher concentration of CaO but they differ from the CGPAR01 individuals. The Marnoso-Arenacea del Belice formation, present in the area of Partinico, is excluded as a possible source of raw materials as being highly calcareous, in addition to other differences in composition [40]. Therefore, the area of Partinico would need further investigation to locate possible raw material sources. In this group, only sugar cones are present (Tables S2 and S3).

At the right side of the dendrogram (Figure 6), calcareous individuals with the highest CaO content (12–15 $w$%) show a structure divided into two groups along with three isolated individuals. The latter correspond to the underwater recovery (UND001), one of Palermo's geological deposits (DPAM01), and an individual (STE004) from the site of Palazzo Steri–Chiaramonte (Palermo). Another ungrouped individual can be found on the extreme left of the dendrogram and corresponds to the other geological deposits from Palermo (DPAM02). Some individuals from Palermo join in a small group, labelled CGPAL01, composed of four sugar cones from Convento di San Giovanni at Baida (BAD) and two molasses jars from Castello della Favara a Maredolce (MAR). Finally, the largest group plotted on the right, labelled CGPAL02, includes 14 individuals recovered from Palazzo Steri–Chiaramonte (STE), eight from Castello della Favara a Maredolce (MAR), and one from Convento di Baida (BAD), all at Palermo; it also includes two individuals from Himera (HIM174–HIM175) and six from Partinico (PAR). Sugar cones, molasses jars, noria vessels, and cantaros are present in this large group (Tables S2 and S3). Moreover, in this case, PE shows the grouping of these individuals in one fabric, PPAL01 (Supplementary Material 2), although with an internal variability, as also seen in the biplots discussed above. PPAL01 is a fine- to medium-grained fabric composed mainly of quartz with a minor frequency of feldspars, mica, and pyroxenes, and a variable presence of a coarser fraction composed of limestone, chert, and sandstones (Figure 9a,b). The coarse fraction is larger and more frequent in subgroup PPAL01b. In most of the individuals grouped in this fabric, voids are filled by micritic inclusions, such as those described by Cau Ontiveros et al. as micritic clots and fringes along pores [76] due to the recrystallisation after firing of residual CaO already present in the paste. This fabric is compatible with the characteristics of Ficarazzi deposits available in the Palermo area [40]. Its chemical and fabric variability, although not matching, may be due to intra- and inter-variability of the clay deposits. Compared to the Ficarazzi formation deposits previously characterised by Montana et al. [41], the concentration of MnO, Zr, Ba, and Zn is higher in the archaeological individuals. Moreover, the two geological deposits analysed, DPAM01 and DPAM02, differ from the published dataset for the same elements. These two share similar rock fragments with the archaeological material but are also dissimilar regarding different points. DPAM01 is well packed with silt size inclusions, whereas DPAM02 has a finer groundmass with heterogeneously distributed

sand size inclusions (Figure 9e,f). From the chemical perspective, DPAM01 has a lower concentration of many elements, mostly in $Al_2O_3$, Ce, Rb, and Ga, and a high concentration of $SiO_2$ compared to DPAM02. Our archaeological group fits in the middle of these two extremes, both from a fabric and chemical point of view. Nevertheless, DPAM01 was chosen for the mechanical and thermal performance test as the one more resembling the archaeological individuals for the grain size distribution and packing.

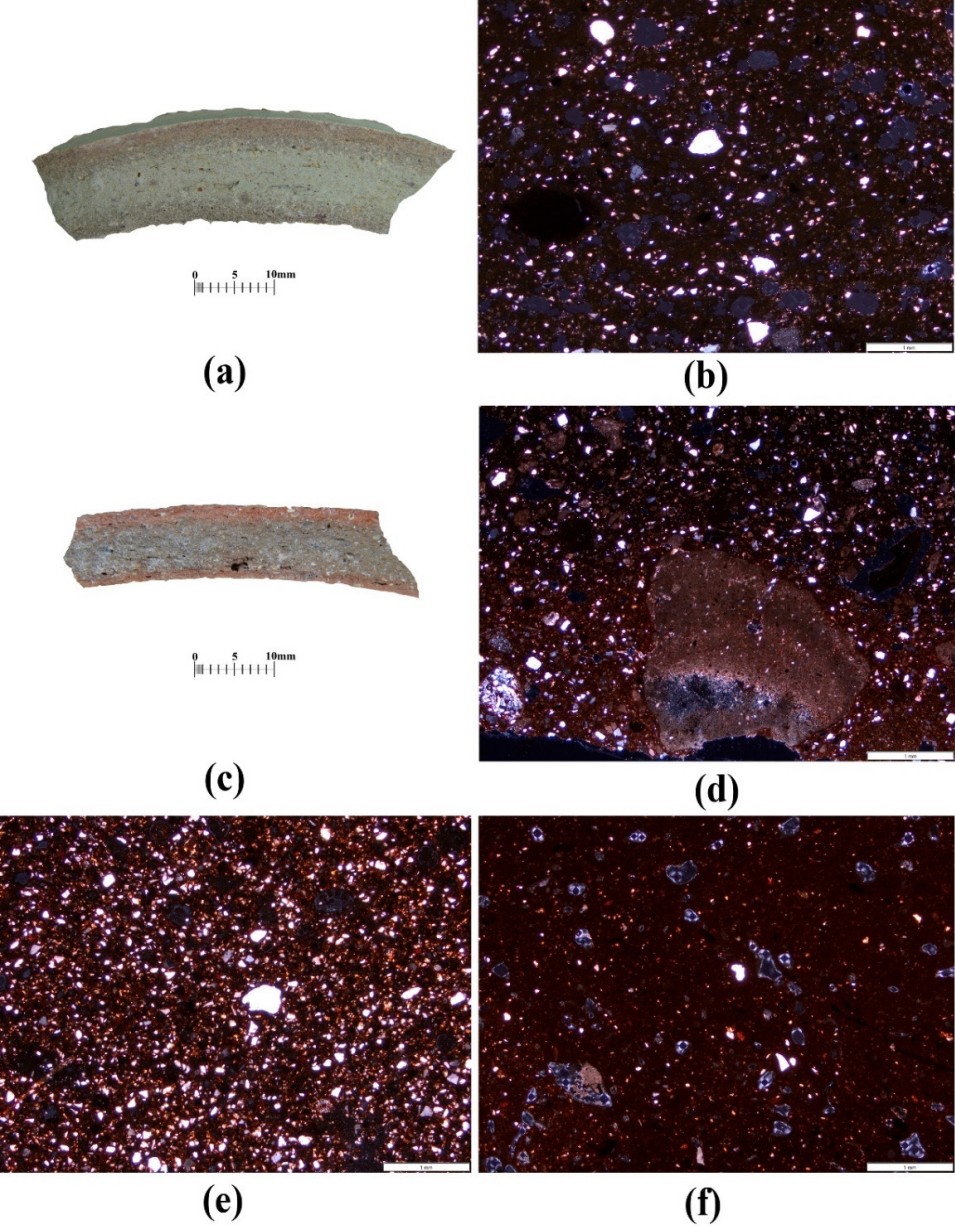

**Figure 9.** Fabric PPAL01a: macrophotograph (**a**) and photomicrograph in XP (**b**) of individual HIM174; fabric PPAL01b: macrophotograph (**c**) and photomicrograph in XP (**d**) of individual BAD007; photomicrograph of briquettes DPAM01 (**e**) and DPAM02 (**f**) fired at 850 °C.

Regarding the isolated individuals, STE004 suffers severe contamination of Pb because of analytical interference caused by Pb, in addition to other chemical elements (Rb, Y, Ce, Th) [89,90]; however, by removing these elements from the statistical treatment, it groups with CGPAM02, and PE confirmed this grouping [91].

PE allows hypotheses to be drawn regarding formation [69,70], although macroscopic observations are, in this case, most suited due to the large size of these vessels. Individuals

of PHIM01-CGHIM01 seem wheel-thrown (Figure 8b), probably in segments; in some cases, the joint of the section is visible by the change in the preferred orientation of voids and inclusions. These features can be better observed macroscopically for the clear traces of a wheel on the body (Figure 10a), and by areas of depressions in the vessel's body (so-called Y marks) [63] on the possible junction with the rim part (Figure 10c). Moreover, the bottom part of the cones shows a variation of thickness towards the hole, probably due to the fact that the bottom part was thrown upside down (Figure 10b). External cordons seem rolled up and outwards from the rim (Figure 10d). Strong orientations of voids and inclusions that may be due to wheel throwing were found in a few individuals grouped in fabric PPAL01 (especially BAD001-007), but the majority show no to poor preferred orientation of voids and inclusions. Moreover, individuals in PPAR01 show poor orientation of voids and inclusions. Macroscopically, these individuals show uneven wall thicknesses along the two sides of the cones, in addition to traces of the wheel on the surface, suggesting the use of hand-forming methods and possibly rotary devices only for finishing (Figure 10d,e); holes are not always centred and also here there are signs of the bottom part being added (Figure 10f), probably upside down as a variation in thickness towards the hole can be observed. A systematic study of the forming traces coupled with experimental reproduction will support these initial hypotheses.

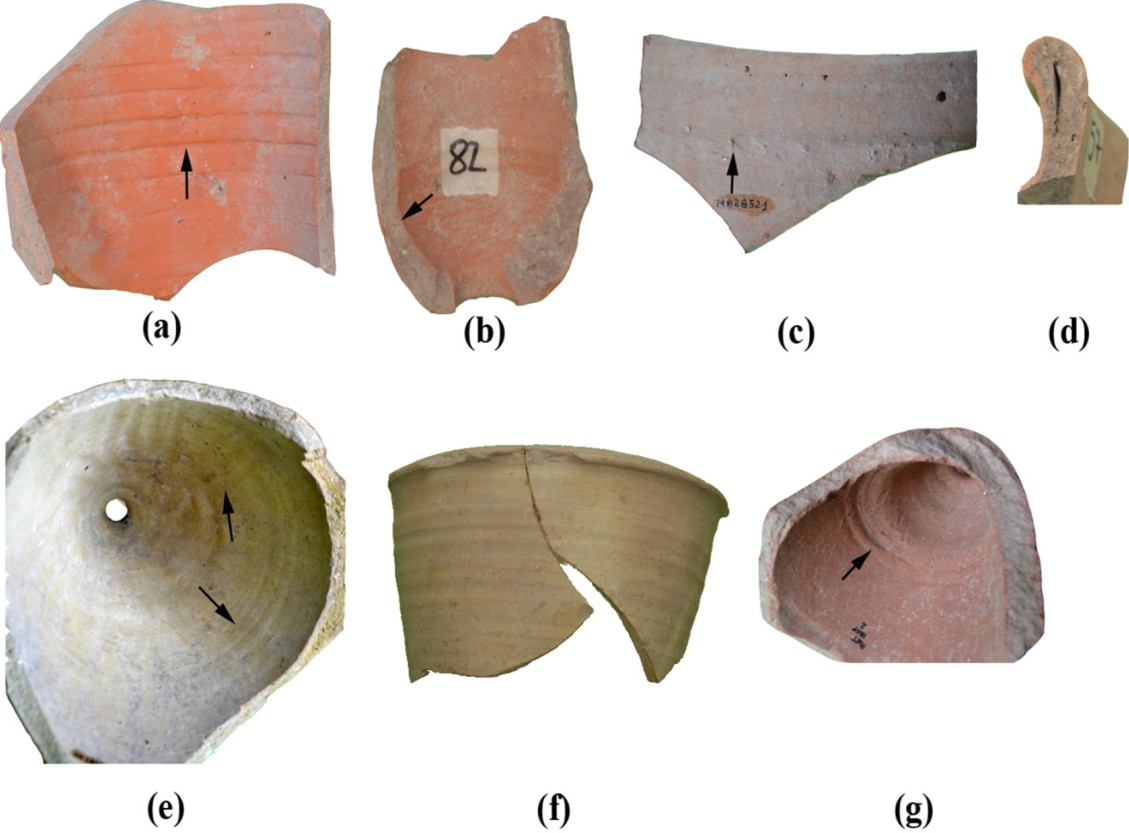

**Figure 10.** Some examples of the forming macro-traces observed on the sugar cones: (**a**) body of an individual from Himera (not numbered), rings produced by the wheel; (**b**) bottom part of HIM082, change in wall thickness towards the hole; (**c**) inside part of the upper part of HIM172, Y marks; (**d**) section of the rim of HIM057, rolling and folding of the clay outwardly; (**e**) inside bottom part of HIM175, wheel ring, and traces of finger pulling clay towards the hole; (**f**) outside upper part of HIM175 showing wheel rings; (**g**) inside bottom part of PAR005, change in thickness towards the hole, and joint of two sections of the vessel. Arrows indicate points where the features described can be observed. Not to scale.

Regarding firing strategies, PE suggests that most of the individuals from PHIM01 and PPAM01 show signs of being fired at a high temperature according to the micro-mass optical activity. Individuals in PPAR01 seem to have been fired at a low temperature and in oxidising conditions. Individuals of each group were mineralogically and microstructurally characterised by means of XRD and SEM to verify these initial observations.

In order to contribute to the knowledge of technical aspects of the production of the artefacts considered in this study, chemical results show that the analysed individuals are ceramics considered as being calcareous ($5–6\% < CaO < 20–25\%$). Accordingly, calcareous ceramics commonly develop high-temperature phases and a light microstructure with a progressive formation of a vitreous phase [65,92,93]. The ceramic phase triangle ($CaO + Fe_2O_3 + MgO$)-$Al_2O_3$-$SiO_2$ (Figure 11) shows how all the individuals analysed in this study are positioned in the quartz–anorthite–wollastonite triangle, which is characteristic of calcareous ceramics. With this in mind, mineralogical and microstructural characterisation is explained below based on the chemical and petrographic results. The complete set of XRD diffractograms and SEM photomicrographs was previously published [94,95].

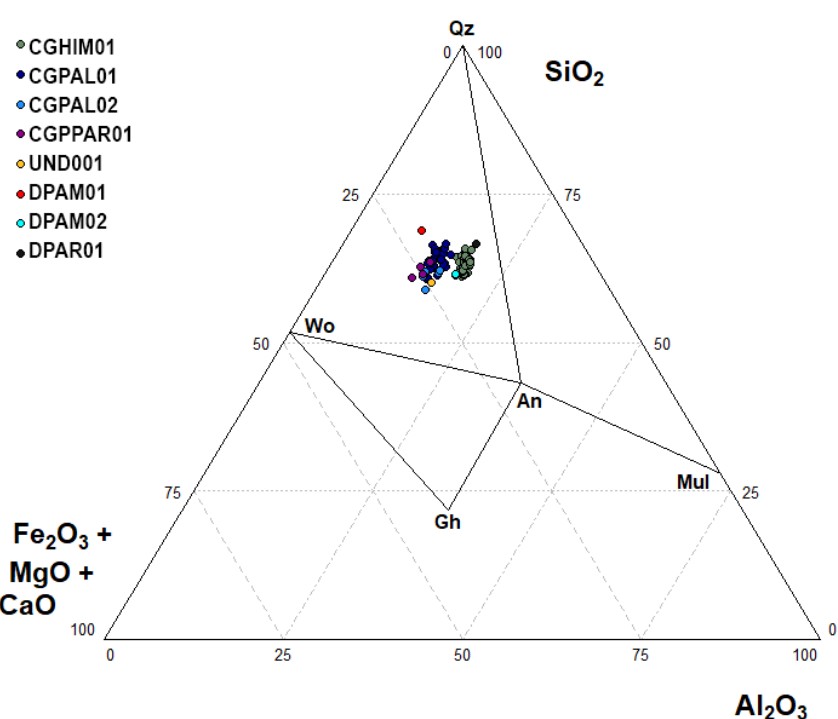

**Figure 11.** Ternary diagram of the system ($CaO + Fe_2O_3 + MgO$)-$Al_2O_3$-$SiO_2$. An: anorthite ($Ca[Al_2Si_2O_8]$); Gh: gehlenite ($Ca2Al[AlSiO_7]$); Mul: mullite ($Al_6[Si_2O_{13}]$); Qz: quartz ($SiO_2$); Wo: wollastonite ($CaSiO_3$) (abbreviations according to [96]).

The study of the XRD diffractograms of the 47 individuals of the CGHIM01 chemical group allowed the identification of four fabrics (F1 to F4), i.e., different categories of association of crystalline phases, representing four different equivalent firing temperatures (EFT) (Table 4). F1 (HIM001, 015, 070, 99, and 121) presents the three characteristic peaks of illite-muscovite at lower angles, quartz, plagioclase, alkali feldspar, an intense peak of calcite, and hematite (Figure 12a, HIM099). Amongst these phases, only the presence of the three illite-muscovite peaks, which are usually present in EFT up to 950–1000 °C,

points to an EFT below this range for F1. Hematite is observed in individuals of all fabrics and therefore fails to provide indications to estimate the EFT. The microstructural study of HIM099 shows a generally no-vitrification (NV) microstructure with a few areas on the margins that are starting the vitrification process (Figure 13a). These observations allow us to estimate an EFT of below 800 °C for these individuals. Diffractograms of F2 (HIM036, 042, 044, 047, 080, 091, 098, 135, 142, 146, and 158) show a decrease in the peaks of calcite, a clear development of plagioclase, and the development of gehlenite and initial peaks of pyroxene, firing phases that crystallise above 800 or 850 °C. The three peaks of illite-muscovite are still present, which indicates an EFT in the range of (850–950/1000) °C (Figure 12a, HIM091). Two individuals were examined by SEM from this group, HIM091 and 142, revealing an extensive vitrification stage (Vc) (Figure 13b). The EFT may be in the range proposed. F3 (HIM013, 21, 35, 50, 75, 112, 148, 176 and 177) shows the almost complete decomposition of illite-muscovite, which preserves only one peak visible for most of the cases, and of calcite, which is completely decomposed or showing a very reduced peak. On the other side, pyroxene, plagioclase, and alkali feldspar are now well-developed phases. Gehlenite is still present but with significantly reduced peaks (Figure 12a, HIM075). All this evidence enables us to estimate an EFT of around 950 °C. SEM examination of individuals HIM035 and 075 reveals a well-developed vitrified microstructure with many areas having micro- and macro-pores (Figure 13c). According to this microstructure, the EFT is estimated at around 950–1000 °C. Finally, illite-muscovite and calcite completely disappear in the diffractograms of fabric F4, and gehlenite peaks are greatly reduced or decomposed for most of the cases (HIM003, 005, 7, 8, 18, 19, 23, 29, 31, 34, 45, 55, 56, 57, 60, 62, 71, 72, 82, 84, 92, 109, 113, 147, 167, and 178). In addition, an important development of plagioclase, hematite, and pyroxene is observed (Figure 12a, HIM178). These characteristics indicate an EFT above 950/1000 °C. From SEM, both HIM031 and 178 showed a total vitrified microstructure with macro-pores (Figure 13d); this suggests a higher EFT than the previous groups, probably in the range of 1000–1100 °C. Sugar cones and molasses jars can be found in all the fabrics but most of the molasses jars (13 over 17) are included in the highest EFT fabric, F4.

**Table 4.** Table of the estimated equivalent firing temperature (EFT) according to X-ray diffraction (XRD) fabrics and vitrification stages by scanning electron microscopy (SEM, in bold individuals, examined). Afs: alkali feldspar; Cal: calcite; Px: pyroxene; Gh: gehlenite; Hem: hematite; Ilt: illite-muscovite; Pl: plagioclase; Qz: quartz; Spl: spinel (abbreviations according to Whitney and Evans, 2010). NV: no vitrification; IV: initial vitrification; Vc: extensive vitrification; TV: total vitrification. low calc.: low calcareous, calc: calcareous.

| XRF Group | Petrographic Fabric | XRD Fabric | Individuals | Vitrification Stage Core/Margins | Calcareous/Low Calcareous | EFT °C |
|---|---|---|---|---|---|---|
| UND001 | PPAL01 | F1—Afs, Cal, Mg-Cal, Hem, Pl, Px, Qz, Spl | UND001 | / | low calc. | >950–1000 |
| CGHIM01 | PHIM01 | F1—Afs, Cal, Hem, Ilt, Pl, Qz | HIM001, 015, 070, **099**, 121 | NV | calc. | <800 |
| | | F2—Cal, Px, Hem, Ilt, Gh, Pl, Qz | HIM036, 042, 044, 047, 080, **091**, 098, 135, **142**, 146, 158 | Vc | calc. | (850–950/1000) |
| | | F3—Afs, Di, Hem, Ilt (1-2/3), Gh, Pl, Qz | HIM013, 021, **035**, 050, **075**, 112, 148, 176, 177 | Vc+ | calc. | (950/1000) |
| | | F4—Afs, Cal, Gh, Px, Hem, Pl, Qz | HIM003, 005, 007, 008, 018, 019, 023, 029, **031**, 034, 045, 055, 056, 057, 060, 062, 071, 072, 082, 084, 092, 109, 113, 147, 167, **178** | TV | calc. | (1000–1100) |

**Table 4.** *Cont.*

| XRF Group | Petrographic Fabric | XRD Fabric | Individuals | Vitrification Stage Core/Margins | Calcareous/Low Calcareous | EFT °C |
|---|---|---|---|---|---|---|
| CGPAR01 | PPAR01 | F1—Cal, Ilt, Pl, Qz | PAR**002**, 003, 006, 009 | NV | calc. | <800 |
| CGPAL01 | PPAL01 | F1—Cal, Hem, Ilt (2/3), Gh, Pl, Qz | BAD001, 02, **006**, **007** MAR008 | Vc | calc. | (950–1000) |
|  |  | F2—Cal, Px, Hem, Pl, Qz | **MAR006** | TV | calc. | (1000–1100) |
| CGPAL02 | PPAL01 | F1—Cal, Ilt (1/3), Pl, Qz | **STE011**, 013, **014**, PAR005 | NV/IV | calc. | 800 |
|  |  | F2—Cal, Ilt (1/3), Hem, Pl, Qz | STE003, MAR005, **PAR004** | NV/IV | calc. | 800 |
|  |  | F3—Cal, Gh, Ilt (1/3), Hem, Pl, Qz | STE012, PAR007, 008, 010, **011** | V | calc. | (950–1000) |
|  |  | F4—Afs, Cal, Px, Gh, Ilt (1/3), Hem, Pl, Qz | STE001, 002, 005, 006, 008, 009, 016, **BAD004**, **HIM174**, 175 | TV | calc. | (1000–1100) |
|  |  | F5—Afs, Cal, Px Ilt (1/3), Hem, Pl, Qz | **MAR009** MAR001, 004, 007, STE004 | TV | calc. | (1000–1100) |
|  |  | F6—Afs, Cal, Px, Hem, Pl, Qz | MAR003 and MAR010 | / | calc. | (1000–1100) |
|  |  | F7—Cal, Px, Hem, Pl, Qz, Gh | **STE010** | TV | calc. | (1000–1100) |
|  |  | F8—Afs, Di, Pl, Qz | **MAR002** | TV+ | calc. | (1000–1100) |

The four individuals belonging to the CGPAR01 group are four sugar cones that exhibit peaks of illite-muscovite, quartz, calcite, and plagioclase (Figure 12b) (Table 4). No firing phases are present, which would indicate an EFT below 800 °C. SEM examination of PAR002 confirms this interpretation by showing an NV microstructure (Figure 13e). In spite of the small number of individuals from this group, a low temperature can be characteristic of these products.

Regarding the six individuals in CGPAL01, the examination of the diffractograms allowed the identification of two fabrics (Table 4). F1, which includes most of the individuals (BAD001, 002, 006, 007, and MAR008), exhibits two of the three peaks of illite-muscovite at lower angles, calcite, quartz, plagioclase, hematite, and gehlenite and, in some instances, the low-intensity peaks of pyroxene are observed (Figure 12c). By SEM, BAD006 and 007 show a Vc core with micro-bloating in a few areas at the core of the section (Figure 13f). The only individual corresponding to F2 (MAR006) shows the total decomposition of illite-muscovite, the decrease in the gehlenite peak, and development of pyroxenes; the high peak of calcite is due to post-firing crystallisation, as was verified by PE (Figure 12c). According to these results, an EFT in the range of 950–1000 °C can be proposed for F1 and around 1000–1100 °C for F2. SEM examination of MAR006 shows a TV microstructure and confirms the higher temperature proposed for F2. Therefore, it would seem that high temperatures would be the option preferred by the potters in this case, for both sugar cones and molasses jars.

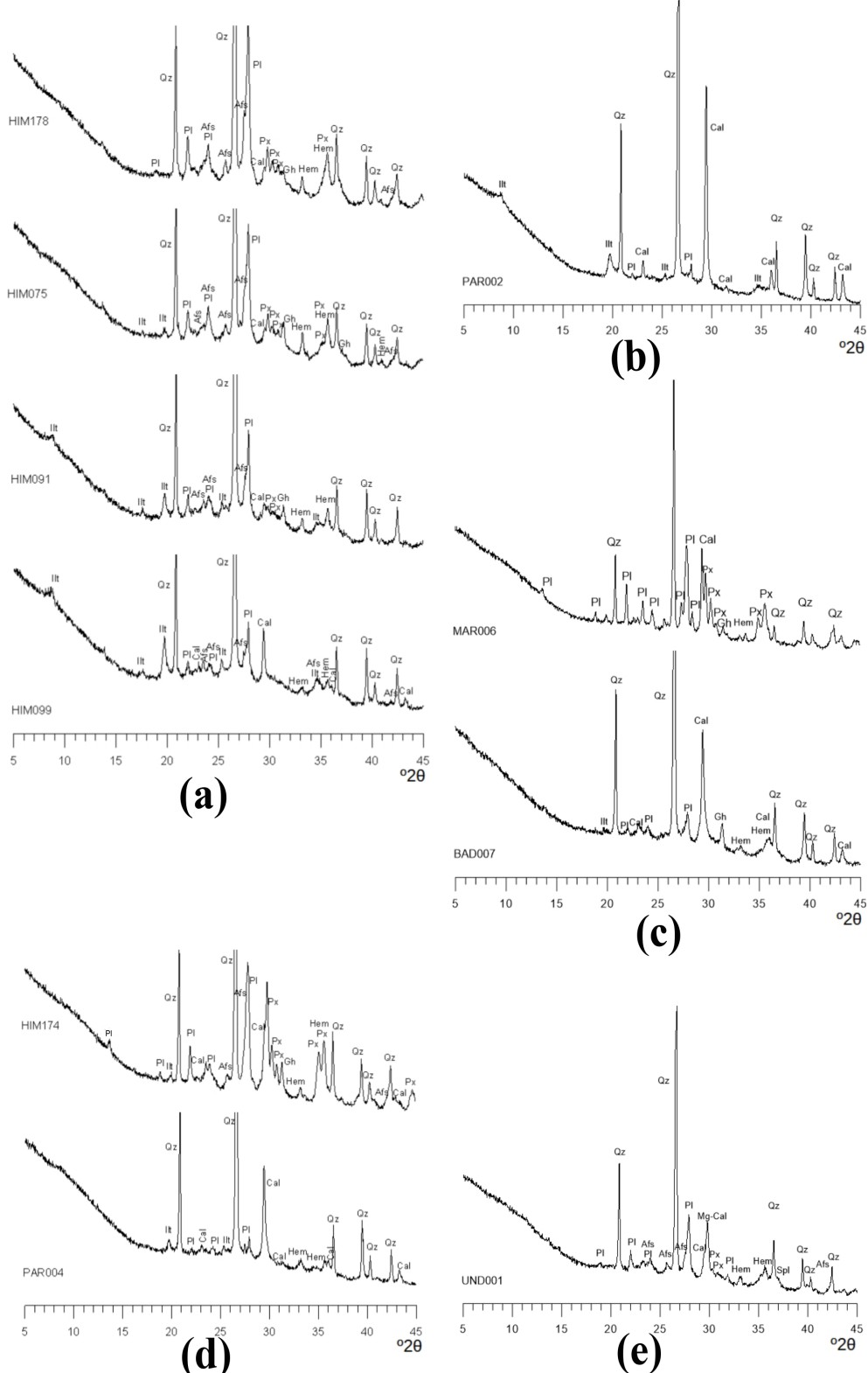

**Figure 12.** Diffractograms of some of the archaeological individuals discussed by the chemical groups. (**a**) CGHIM01: individual HIM099, F1; individual HIM091, F2; individual HIM075, F3; individual HIM178, F4. (**b**) CGPAR01: individual PAR002, F1. (**c**) GCPAL01: individual BAD007, F1; individual MAR006, F2. (**d**) GCPAL02: individual PAR004, F1; individual HIM174, F4; (**e**) individual UND001. Afs: alkali feldspars; Cal: calcite; Gh: gehlenite; Hem: hematite; Ilt: illite-muscovite; Hem: hematite; Mg-Cal: magnesium calcite; Pl: plagioclase; Px: pyroxene; Qz: quartz; Spl: spinel (abbreviations according to [96]).

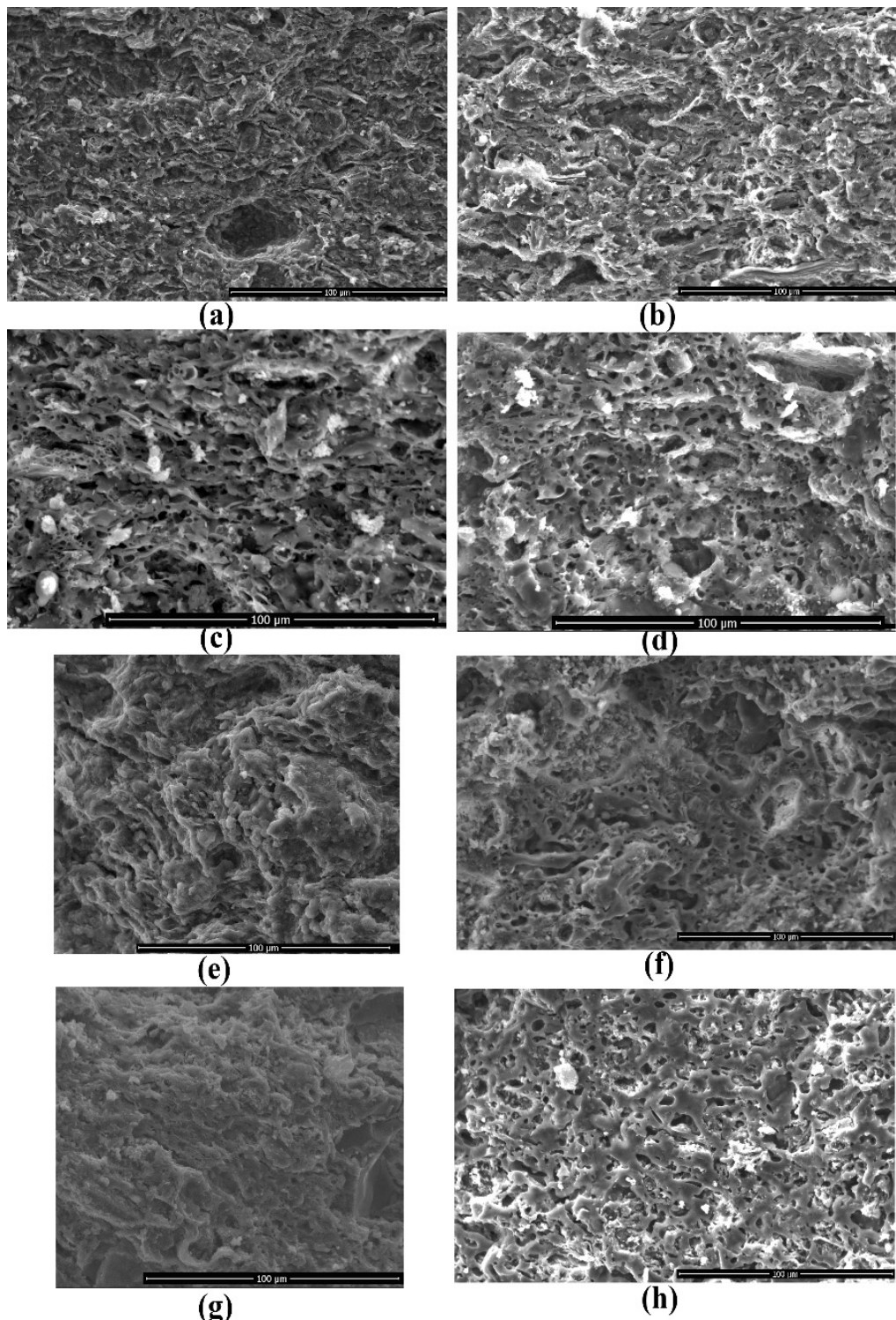

**Figure 13.** SEM photomicrograph (SE) of some of the archaeological individuals discussed. Chemical group CGHIM01: (**a**) HIM099, F1, NV microstructure; (**b**) HIM091, F2, V microstructure; (**c**) HIM075, F3, V+ microstructure; (**d**) HIM178, F4, TV microstructure. Chemical group CGPAR01: (**e**) PAR002, F1, NV microstructure. Chemical group CGPAL01: (**f**) BAD007, F1, V microstructure. Chemical group CGPAL02: (**g**) PAR004, F2, IV microstructure; (**h**) HIM174, F4, TV microstructure. Photomicrograph taken at ×2000, except for (**c**,**d**) at ×1000 taken with a JEOL J6510.

The second group related to Palermo, CGPAL02, is the second-largest group identified in this study, formed by 31 individuals. Their diffractograms show up to eight fabrics

indicating a wide range of EFT from <800 °C to >950/1000 °C (Table 4). The F1 (STE011, 013, 014, PAR005) and F2 (STE003, MAR005 and PAR004) exhibit the same mineral phases: illite-muscovite, quartz, calcite, and plagioclase, with the exception of hematite, which is only present in F2 (Figure 12d, PAR004). In both cases, only the illite-muscovite peak of 4.5 Å is visible, a fact that might lead us to think about the decomposition of primary minerals of clay and calcite and the crystallisation of firing phases. According to the mineralogical characterisation of other Ficarazzi formation deposits [41], these are not as rich in illite-muscovite as in other clays and therefore the development of illite-muscovite peaks may not be an explanation for firing temperature changes. In addition, the presence of the peak of calcite, which is prominent in both fabrics, cannot be used for the estimation of a low EFT; PE indicates that almost all these individuals show the presence of calcite recrystallised after firing. In the samples examined by SEM from these two fabrics, a similar microstructure can be observed: STE011 and 014 are generally NV but with a few spots showing IV; PAR004 shows an NV core and IV margins (Figure 13g). According to these observations, an EFT ~800 °C is estimated. The next three fabrics, F3 (STE012, PAR007, 008, 010, and 011), F4 (STE001, 002, 005, 006, 008, 009, 016, BAD004, MAR001, 004, 007, and HIM174, 175), and F5 (MAR009) still show the peak of 4.5 Å of illite-muscovite, but now the clear crystallisation of gehlenite is observed for F3 together with an important development of plagioclase and hematite. Some individuals present an initial development of pyroxenes, clear for F4 (Figure 12d, HIM174) and more developed for F5, a fabric in which gehlenite is not present. All the individuals from these three fabrics present a peak of calcite to a greater or lesser extent. Consequently, an EFT >850 °C is clear for the three fabrics and an EFT around 950/1000 °C can be estimated for F4 and F5. SEM examination allows this temperature to be increased to the 950–1100 °C range; PAR011 from F3 shows a Vc microstructure with some areas at the core showing an advanced state of Vc$^+$; both individuals from F4 (BAD004, HIM174) show dense vitreous masses (TV, Figure 13h), whereas individual MAR009 has an over-vitrified microstructure. Finally, three more fabrics were identified: F6 (MAR003 and MAR010), F7 (STE010), and F8 (MAR002), which would be clearly fired above 950/1000 °C. None of the individuals from these fabrics show the presence of the illite-muscovite peaks; the peak of gehlenite is absent in F6 and F8, and the peak of calcite is completely decomposed in F8. The microstructure of STE010 is characterised by total vitrification, whereas in the MAR002 the microstructure is very disturbed but areas totally vitrified are recognisable. The variety of microstructures and XRD fabrics encountered in this group does not seem correlated to ceramic shape or archaeological context. Only the five molasses jars and the three *noria* vessels of this group are included in F3 and F4, which are the products fired at a high temperature, as was observed for Himera; however, the number of these is too small compared to that of sugar cones to consider this result significant.

The underwater individual UND001 was included in fabric PPAM01 (Supplementary Material 2), but chemically it seems that it does not match the other individuals from CGPAL01-02. Compared to these groups, UND001 shows clear chemical changes that typically are found in sherds from marine environments [97–105]: a high concentration of MgO, which often occurs in underwater findings, causing a decrease in CaO, although in this case, CaO is still high (around 10 $w$%). The diffractogram of UND001 shows the absence of hydrotalcite, a mineral phase reported as usually developing in seawater environment, but the presence of magnesium calcite has also reported as being formed in the same environment; a peak of spinel is also evident (Figure 12e). This suggests that the significant amounts of both MgO and CaO are allochthonous, and fixed by secondary phases, and that UND001 is a low calcareous individual. In addition to the CaO and MgO content, UND001 has strong similarities to other CGPAL01-02 individuals, and when removing MgO and CaO from the statistical treatment of the data, it fits within the CGPAL02 group. However, as CGPAL01-02 individuals usually bear higher CaO content, UND001 is considered a loner at present. The firing temperature of this individual is probably in the higher range due to the absence of any of the illite-muscovite peaks and the presence of spinel. The pyroxene peak is probably due to its primary presence in the paste.

Based on the chemical and petrographic results, two geological deposits were chosen to further perform mechanical and thermal tests on experimental briquettes fired at different temperatures (Figure 14). DHIM01 perfectly matches the archaeological individuals in group CGHIM01, whereas DPAM01 was considered the closest to the individuals in groups CGPAL01-02. DPAM01 is the closest to the CaO content, which influences the microstructure development, and the packing and grain size of inclusions that also affect the thermal and mechanical properties. SEM examination of these experimental briquettes allows the consideration of the development of the microstructure of these fabrics at different temperatures (Figure 15). In both DHIM01 and DPAM01, a major change occurs between 950 and 1100 °C when areas with a denser mass are formed. In DPAM01, one can see some areas where macro- and micro-pores are still present, whereas in DHIM the latter prevail. The difference in CaO content between the two deposits (around 7 and 14 $w$%, respectively) is the cause of this different development. Comparing microstructures, a good correspondence between the experimental briquettes and those of the archaeological ceramics is observed.

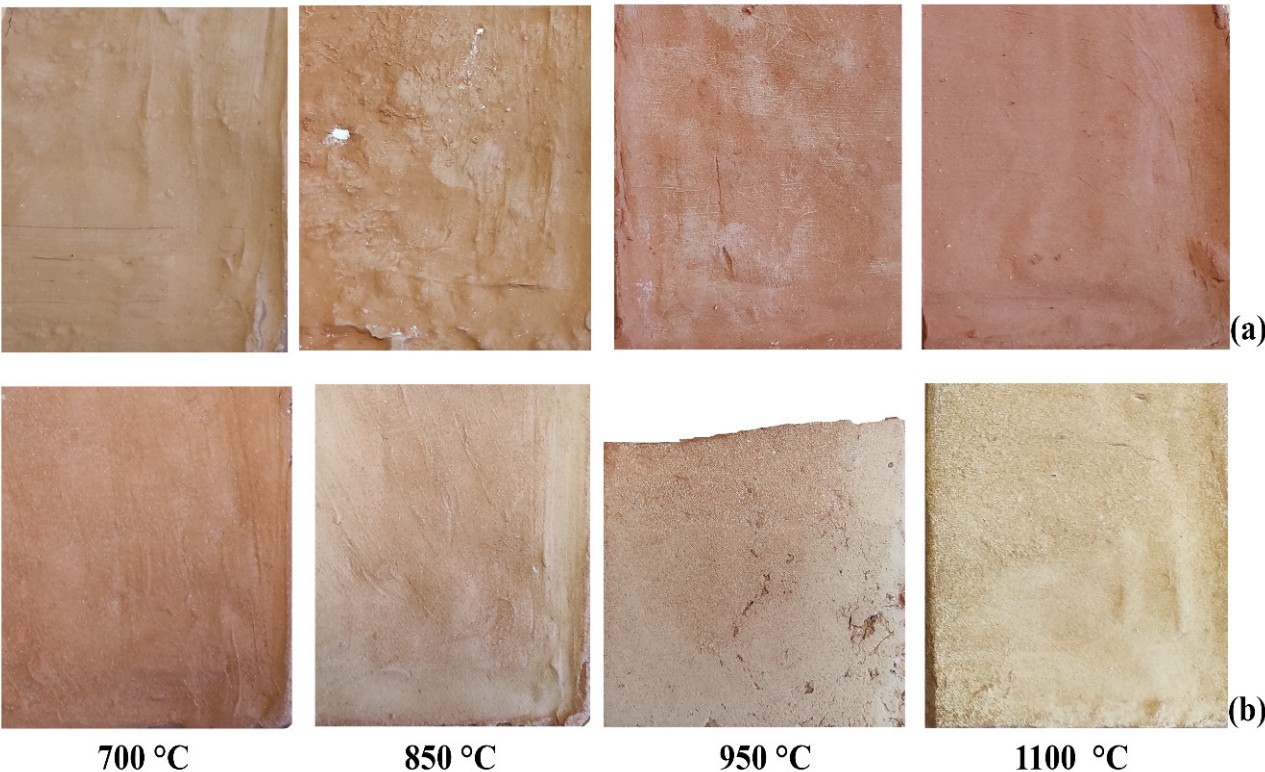

**Figure 14.** Experimental briquettes of DHIM01 (**a**) and DPAM01 (**b**) fired at different temperatures.

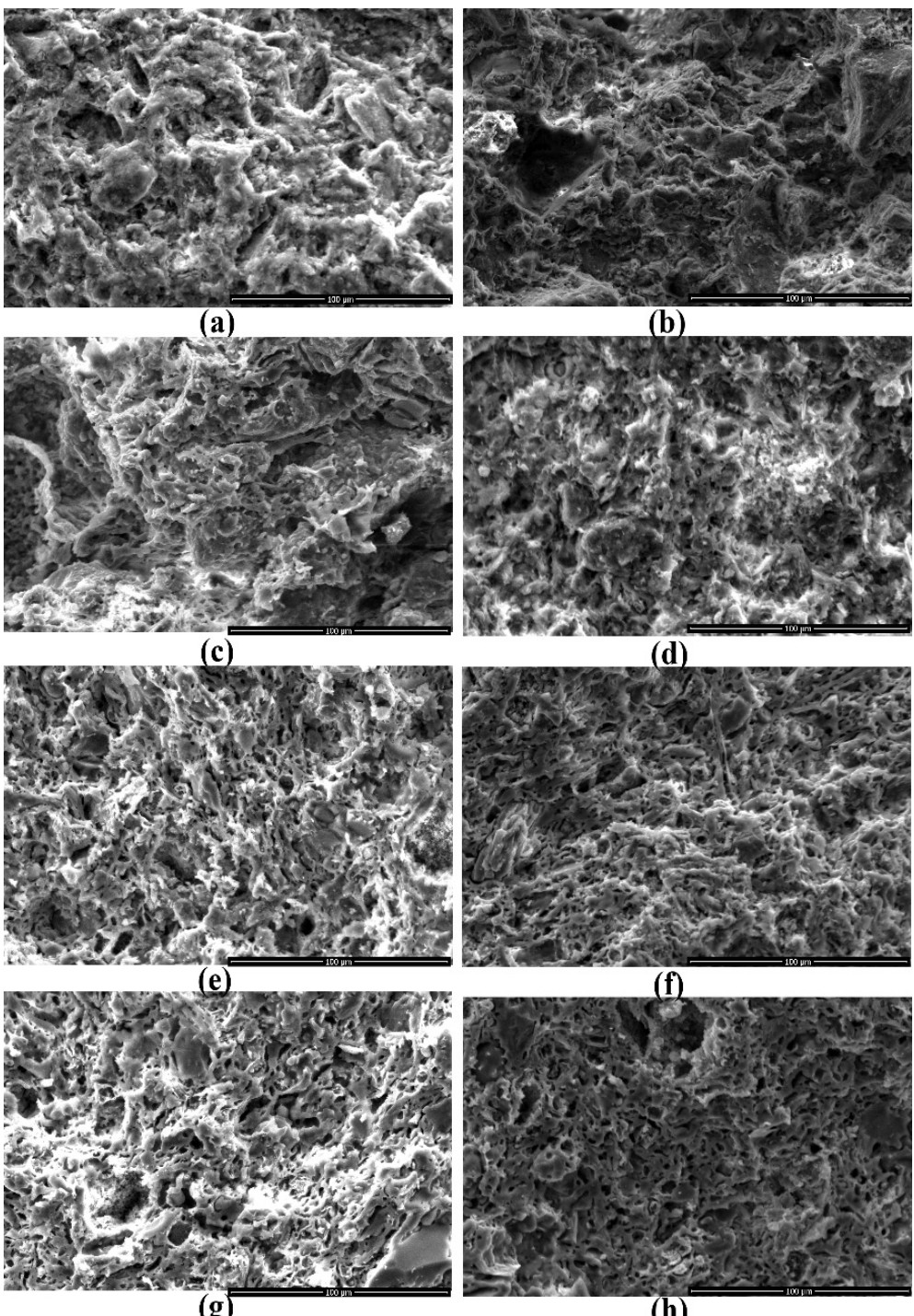

**Figure 15.** SEM photomicrographs (SE) of the microstructure developed by the experimental briquettes of the two geological deposits sampled: (**a**) DHIM01 and (**b**) DPAM01 fired at 700 °C showing an NV microstructure; (**c**) DHIM01 and (**d**) DPAM01 fired at 850 °C, IV microstructure; (**e**) DHIM01 and (**f**) DPAM01 fired at 950 °C, Vc microstructure; (**g**) DHIM01 and (**h**) DPAM01 fired at 1100 °C, TV microstructure. Magnification: ×2000.

In terms of surface treatment, sugar pots present a heterogeneously distributed white appearance, which is clearer when examining complete vessels (Figure 16a,b). Macroscopic examination of the surface and section suggests that vessels were not slipped with a calcareous material; the white area is found in patches on the surface, and in the section the white is very thin, almost invisible (Figure 16c) or fading into the margins. SEM exami-

nation confirmed this hypothesis because no significant microstructural or compositional differences could be detected between the body and the surface (Figure 16d). A similar heterogeneous white layer was observed in the experimental briquettes of DPAM01, visible only on those fired at 850 and 950 °C (Figure 14b). In the archaeological materials, however, it is not clearly related to temperatures because it could be observed on individuals fired at low to high temperatures.

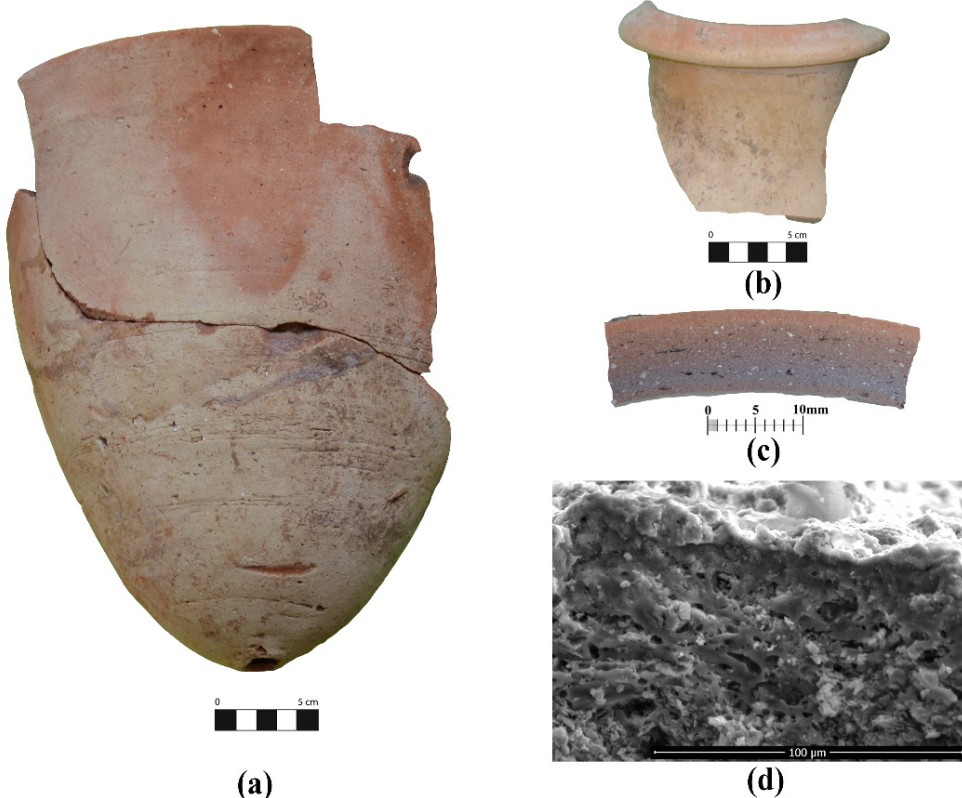

**Figure 16.** (**a**) Sugar cone MAR005 showing heterogeneously distributed white patches on the surface; molasses jar HIM031: (**b**) showing similar white patches on the rim and body, (**c**) in section showing a very thin layer of whitish colour on the surface, (**d**) SEM photomicrograph showing the surface and the body (×1000, SE).

*4.2. Mechanical and Thermal Properties*

4.2.1. Heat Transfer Properties

As expected, the fired geological deposit samples presented thermal conductivities correlated with the firing temperature (Figure 17). This can be explained by the densification of the ceramic matrix during firing and the increasing degree of vitrification (Figure 15) [19]. The specimens of the fired geological deposit sampled close to Himera–Buonfornello (DHIM01), however, presented higher thermal conductivity compared to the fired geological deposit from Palermo (DPAM01), particularly at firing temperatures of above 950 °C. This might be related to the higher CaO content in the paste from Palermo, which apparently results in a higher porosity of the ceramic fabric. The two archaeological individuals measured, HIM178 (CGHIM01, F4) and BAD007 (CGPAL01, F1) presented thermal conductivity values at a level comparable to that of the Himera clay (DHIM01) fired at 1100 °C. Although the EFT of BAD007 is estimated to be lower on the basis of the mineral phases developed, the micromorphology observed under the SEM of both the archaeological samples (Figure 13d,f) resembles the micromorphology of the Himera clay fired at 1100 °C. The thermal diffusivity, which describes the rate of heat transfer, indicates a similar correlation with firing temperature, although in the case of the clay from Palermo (DPAM01) the specimens fired at 950 and 1100 °C are at the same level as the fired clay

from Himera (DHIM01). Hence, ceramics fired from these two clays will transfer heat at the same rate, which concerns, for example, the capability to withstand thermal stress and thermal shock, or the rate at which the content of the vessel exchanges heat with the environment.

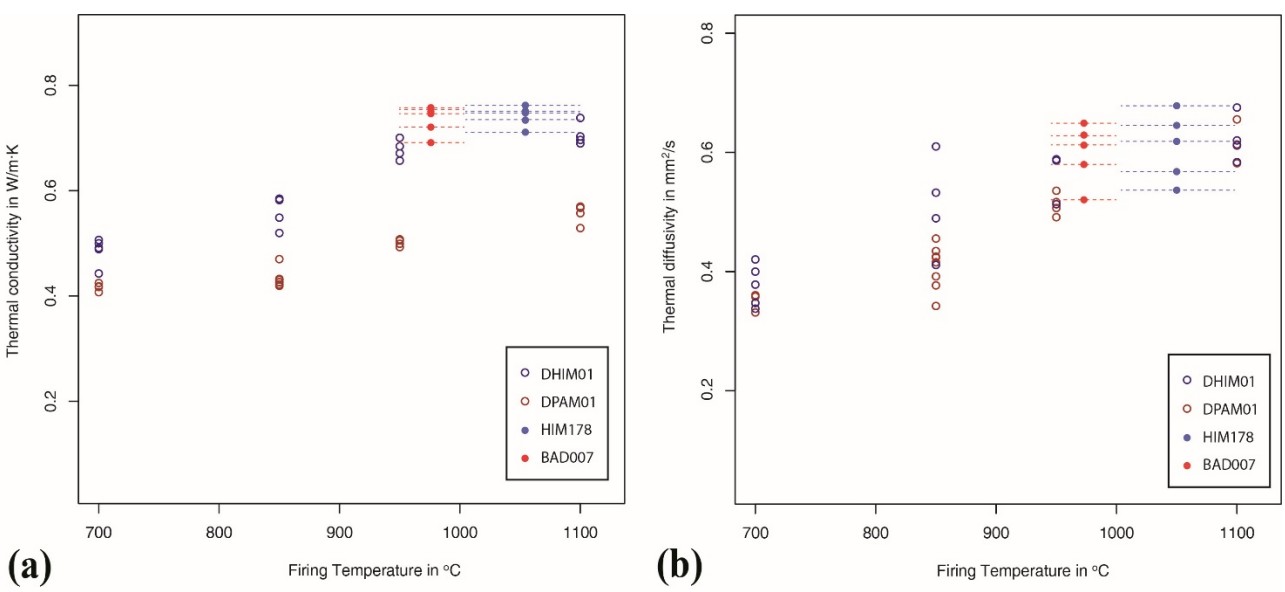

**Figure 17.** Thermal conductivity (**a**) and thermal diffusivity (**b**) measured in the fired geological specimens and in the two archaeological individuals HIM178 and BAD007: the values are plotted versus firing temperature and the EFT estimated for the two archaeological individuals. Raw data are available in [106].

### 4.2.2. Fracture Strength

The fracture strength measured in biaxial flexure tests and three-point bending tests also indicates, as expected, a correlation with firing temperature (Figure 18). Particularly in the case of the Himera clay (DHIM01) a significant increase at firing temperatures from 700 to 850 °C can be observed, which can be explained with the development of the micromorphology and the increase in vitrification in the ceramic fabric [20]. The geological deposit sampled at Palermo (DPAM01) presents comparably lower fracture strength, probably due to the higher content of non-plastic inclusions observed, which introduce flaws and imperfections in the ceramic matrix [21]. By comparison, the recorded load-displacement curves indicated a potentially increased toughness, which can be expected based on the coarser microstructure [20]. The archaeological specimen from Himera HIM178 presents increased fracture strength even in comparison with the Himera clay specimen (DHIM01) fired at 1100 °C. For this, an efficient clay paste preparation and refinement can be assumed before this clay was possibly used for ceramic manufacture. The same may apply to the archaeological specimen from Palermo (BAD007). Although it presents a lower fracture strength than the fired clay specimens from Himera, its fracture strength is increased compared to the fired clay specimens sampled at Palermo.

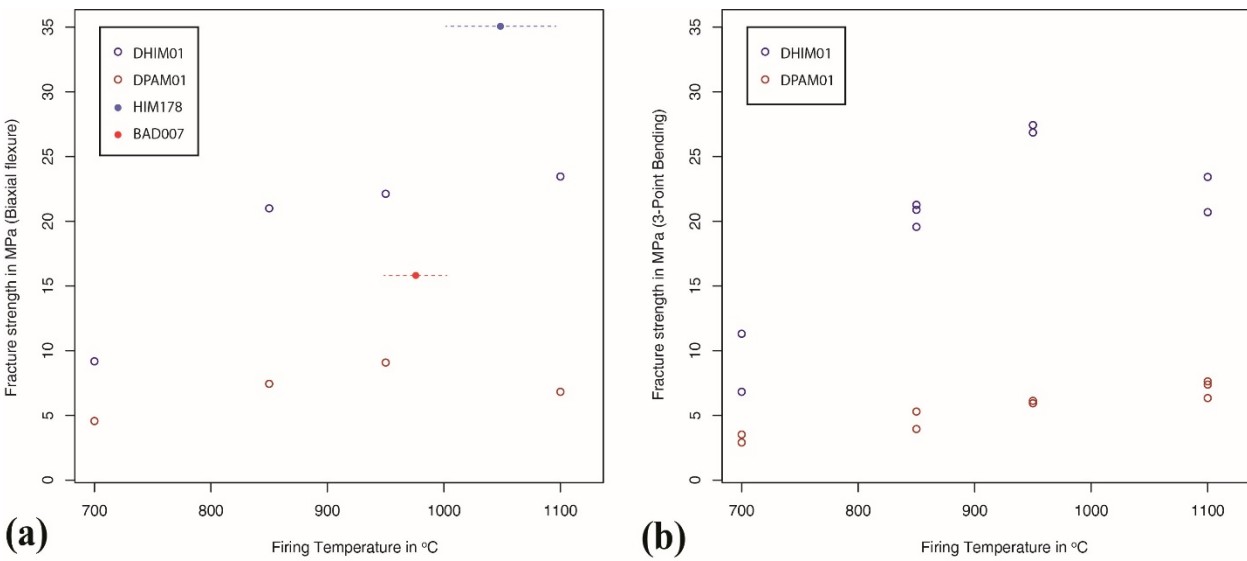

**Figure 18.** Fracture strength measured in the fired clay specimens and in the two archaeological individuals HIM178 and BAD007: the values are plotted versus firing temperature and the EFT estimated for the two archaeological individuals. The left plot presents the results of the biaxial flexure tests (**a**) and the right plot the results of the 3-point bending tests (**b**). Raw data are available in [106].

## 5. Discussion

### 5.1. Sugar Pots Production Areas and Circulation

The results of the chemical and petrographic analyses of the archaeological and geological deposits at least three production areas to be suggested for sugar pots: Palermo, Himera–Buonfornello, and another that we might provisionally assign to Partinico. Further research is needed on archaeological ceramics and local geology in this last case. Palermo production seems more varied than that of Himera, which is most probably due to two factors. First, the sugar pots from Palermo cover a larger chronological range, at least from the 11th to the beginning of the 16th century A.D.; therefore, a certain variability within the same source is expected [107], considering that all the workshops in the city use this source for this entire period. By comparison, Himera–Buonfornello production is dated within a smaller timescale, from the end of the 15th century to the beginning of the 16th century A.D. For this latter case, we are dealing with a sugar production centre, which probably included a ceramic workshop for sugar cones and other wares in its premises [16,38]. In the second instance, regarding Palermo, none of the contexts examined can be considered a production centre *per se*; the production of sugar probably occurred in the surroundings of Palazzo Steri–Chiaramonte and Castello della Favara a Maredolce, as historical studies confirmed [28] (p. 67) [18] (p. 283), but these were mainly residential areas during the phases under study [30,31,34]. Castello della Favara a Maredolce seems to be have been transformed from a residential agricultural function, probably in the mid-13th century [30]. In a second phase, probably by the 15th century, part of its activity was devoted to sugar production, as a larger quantity of sugar cones and molasses jars were found in contact with the four kilns [108]. The ceramics pertaining to this last phase were not examined in this project, however [16]. In addition, the sugar cones found at Convento di San Giovanni di Baida were reused there as building materials [16]. By not being primary production places, sugar pots were probably produced in different workshops operating within the city of Palermo, and reaching the contexts in our study for consumption or secondary purposes. These differences between the two areas are also visible in the vessel design, which is much varied in terms of dimensions and profiles in Palermo compared to Himera–Buonfornello (Figure 3, [16]). This variability can only in part be explained by differences in chronology, and is related to the context of the production of these vessels.

From this study, it emerges that ceramics needed for sugar production were mainly produced in the surroundings of the sugar production centres. In the cases of Himera–Buonfornello and Partinico, the archival record refers to the construction of kilns by sugar production owners to cut the high transport costs and to facilitate the production of large quantities of the vessels [29] (p. 245) [18] (p. 279). Nevertheless, sugar pots produced in Palermo reached the other two areas. In the case of Partinico, although this study concerned survey material, half of the vessels retrieved there match those of Palermo production. Moreover, in terms of design, these vessels resemble those from Palazzo Steri–Chiaramonte [16]. Similarly, two sugar cones found at Himera–Buonfornello belong to the same typology and, indeed, it was found that they were produced in Palermo. In contrast, the circulation of sugar pots from these sugar production centres to Palermo, one of the main consumption and distribution centres, is missing from the archaeological point of view. On the one hand, sugar pots were re-used in their production and their use as transport vessels is not common. On the other hand, the archival record refers to the transport of sugar within sugar cones, for example, from Carini to Palermo [18] (p. 265). The cone found underwater, which was produced in a yet to be defined location in Sicily, suggests that these vessels were circulating, even over long distances. Further archaeological research may contradict this hypothesis but, at present, the only indication we have from archaeological data is that sugar pots from Palermo reached Partinico and Himera–Buonfornello, but not the opposite. It may be wondered whether these were circulating with sugar or empty, to be filled and returned to Palermo; at present, however, we do not possess enough data to discriminate amongst these two scenarios.

*5.2. Sugar Cones Material Properties across Phases and Contexts*

In terms of raw material choices and manipulation, generally calcareous fine-grained pastes were used. At least for Palermo and Himera–Buonfornello, we do not have evidence of those manipulation processes, such as the addition or removal of rock fragments or organics that would drastically alter the properties of the raw materials available. The calcareous content of the two pastes is, nevertheless, different, which may create differences in the microstructure developed during firing and, therefore, in the final properties of the vessel [65,92,93].

Regarding forming techniques, the present evidence suggests that most of the sugar pots were wheel-thrown. Sugar cones have traces of being wheel-thrown in a different section, that is, first the body, then the rim, and finally the bottom part. By comparison, some of the cones found in Partinico and Castello della Favara a Maredolce, and the two found at Himera–Buonfornello but produced in Palermo, show signs of a combination of hand-building and wheel-forming/finishing techniques. These belong to different phases of sugar production and sites; therefore, it is unclear whether we can assign these differences to an earlier phase of sugar pot production or to a different means of forming.

Regarding surface treatment, the whitening effect on the surface is not obtained by applying a calcareous slip. A known method of obtaining the same effect is by mixing seawater with the paste or by smoothing the surface of the vessel with it as a finishing method; during drying and firing, the water evaporates, causing the migration of sodium chloride to the margins and/or the surface, which causes the whitening [99,109]. This method was used since prehistory in the Near East [109] and in Sicily, at least since the 8th century AD [110]. Rye [111] (pp. 35–36) reports that the use of seawater mitigates the effect of calcite decomposition during firing, also lowering the temperature of vitrification. In the case of the sugar pots from Himera–Buonfornello, the addition of seawater may have led to the production of a white surface, rather than mitigate the presence of calcium-rich inclusions, which are not very common in the paste. In contrast, the paste used in Palermo is richer in calcareous content due to the presence of large limestone inclusions. However, in this case it cannot be confirmed that potters always added seawater; the experimental briquette shows the development of this effect without the addition of seawater, probably because this paste is naturally rich in soluble salt. Rather than adding seawater to produce

this surface effect, potters may have chosen those clay deposits which allowed them to produce the desired final result.

Finally, the study of the mineralogical and microstructural development in the examined individuals allows us to discuss some patterns. An EFT in the range of 950–1100 °C is most commonly encountered in individuals from the Himera–Buonfornello group and is probably the intended one to be reached by the potters. A few sugar cones seem to be fired at lower temperatures, and this may have occurred accidentally or resulted from different firings within the same production area. In the case of Palermo production, a variety of EFTs was reconstructed; most of the individuals were in the high ranges of firing temperature (950–1100 °C) and some were fired at lower temperatures. The molasses jars and the noria vessels seem to be fired at high temperatures; otherwise, however, no clear correlation could be found between the estimated EFT and the site, chronology, or vessel shape. This diversity in firing may have resulted from the presence of several workshops in the production of these artefacts, using a technique probably related to their own way of operating. In addition to the diversity in EFT, the microstructure developed between Palermo and Himera–Buonfornello individuals fired at high temperatures is similar, although larger pores could be observed in Palermo individuals. Conversely, all the individuals from Partinico were fired at low temperatures (EFT below 800 °C). As only four individuals belong to this group, it is unsure whether this may be considered a dissimilarity or just the result of an information gap.

These technological profiles were tested for their mechanical and thermal properties. Our results on experimental and archaeological individuals show that, when individuals are fired at high temperature (>950 °C), similar capabilities to transfer heat and to reduce the liability to thermal stress and thermal shock can be observed. Below this firing temperature, the results from the two pastes diverge more, with the Himera sample showing higher thermal conductivity. Similarly, resistance to crack initiation (mechanical strength) and propagation (toughness) changes as a function of the temperature and the developed microstructure. The paste from Himera shows higher fracture strength. By comparison, the Palermo paste appears to show a tough behaviour, enabling the ceramic fabric to absorb energy even after initial crack development. The frequency and size of inclusion in the Palermo paste surely plays a role in this different behaviour [21].

### 5.3. Sugar Pots in the Context of Ceramic Manufacture in Sicily in the Medieval and Post-Medieval Phases

As mentioned above, the sugar pots examined here belong to different phases. Although detailed knowledge of ceramic production is not available for all of these phases, we can try to situate the sugar pots in their manufacturing context. For the first phase (11–13th century AD), the most detailed study available comes from Testolini's research of the operational sequence of 8–11th century ceramics in Sicily [110]. Her reconstruction suggests that, in the 11th century, the glazed and unglazed wares in Palermo were produced with the local Ficarazzi clay, but then differed in terms of the other steps of the manufacturing sequence, thus creating distinguishable final products; she also referred to the whitening of the surfaces on the same wares [110] (pp. 180–202). Of interest is the case of cooking pots, which are made with the same clay with the addition of chert fragments, but then wheel-thrown or coil-built, and fired at high or low temperatures in an oxidising or reducing atmosphere; each *chaîne opératoire* corresponds to a specific shape which, according to Testolini, is a link to the different workshops manufacturing cooking pots in Palermo [110] (pp. 177–179). Previous works on medieval ceramics confirmed the use of Ficarazzi clay for the production of amphorae [112], glazed and unglazed wares for later phases [113–115], and the addition of chert for cooking pots [113,116]. Giarrusso and Mulone [113] also found that a low calcareous paste, probably of Numidian flysch formation, to which chert was added, was also used for cooking pots. Sugar pots, therefore, were manufactured in Palermo, at least with the same paste used for glazed and unglazed materials, but not cooking wares; the whitening on the surface seems to be present also

in other wares, both glazed and unglazed, and both hand and wheel-forming techniques co-existed. Unfortunately, not much knowledge of production processes is available for the 15–16th century phase in Palermo. In contrast, the spatial organisation of the manufacture of the city has been tackled by many scholars [117–120]. In the 10–12th century phase, the ceramic manufacturing activities were located mainly within the city's urban area and along the river Kemonia, with some extension outside the city near the river Oreto, where the clay was extracted [117,120]. These areas continued to be devoted to ceramic manufacture, even in a later phase (from the end of the 13th century to the 14th century), when a southern part of the city walls also seemed to be dedicated to ceramic manufacture [120]. It is not clear from the archive sources whether there was a specialisation of workshops for the manufacturing of specific wares, but craftspeople working with clay for ceramics or tiles were clustered in this part of the city, and were therefore in close contact [118]. A few *trapeti* in Palermo were also located between the Kalsa and the Albergheria neighbourhoods, on the eastern side of the city, but most of the others were in the western part of the city [18] (p. 283). Although the topography of the ceramic and sugar production areas need further research, at first glance it seems that the two forms of production were not concentrated in the same areas (Figure 19).

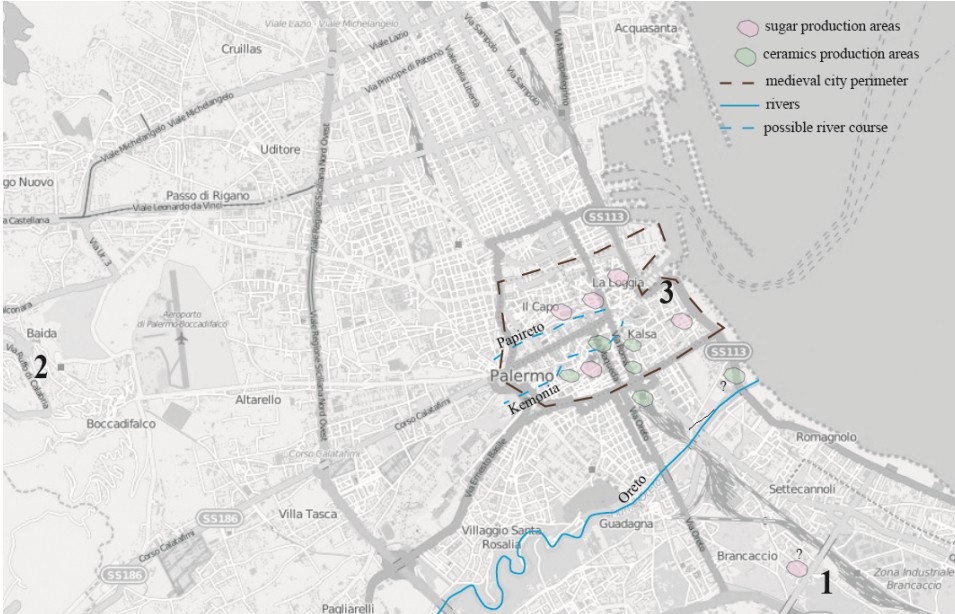

**Figure 19.** Map of Palermo with the sites sampled (1: Castello della Favara in Maredolce, 2: Convento di Baida, 3: Palazzo Steri–Chiaramonte), and the areas of sugar production according to archival sources [18,28] and the areas of ceramics production according to archival and archaeological sources [117–120]. The localisation of the areas has to be considered to be broad.

In contrast, in the case of Himera–Buonfornello, the production of sugar cones took place within the premises of the *trapetum* [29] (p. 245), as was common in this second phase of sugar production on the island, when it spread outside the urban boundaries of Palermo. Termotto [121] describes in detail the organisation and the labour division of the *trapetum* of Galbinogara, one of the biggest sugar production centres, and which was not far from that of the Himera–Buonfornello. He mentions that the clay sources of Collesano, a nearby town, may have been used to produce the sugar pots, which were most probably fired at the *trapetum*. Around this place grew a number of crafts and structures linked with the lives of the workers connected with the sugar production, who were often from other territories and worked seasonally in the *trapetum* [122]. In the case of Himera–Buonfornello, different potters moved their production near the *trapetum*, where the vessels were immediately used and where potters worked in the same restricted area with the other workers of the *trapetum*. Regarding our knowledge of ceramic production in

the area, the use of Terravecchia formation deposits has been attested to since the Greek phases [123,124]. In contrast, for the chronological range, the only available study is that of D'Angelo et al. [125], which indicates that this clay was used for the production of the polychrome glazed Polizzi ware. Kiln wasters of this ware were retrieved together with sugar pots at Himera–Buonfornello, suggesting that the production was performed by the same craftspeople manufacturing the sugar pots. The individuals examined came from Polizzi and not from Himera–Buonfornello, but they share the same type of raw material. However, D'Angelo et al. [125] reported that Polizzi ware was slipped with a different material compared to that used for sugar pots. Future research should examine whether the Polizzi ware vessels made at Himera–Buonfornello share the same characteristics.

*5.4. One Function, Multiple Choices*

In the reconstructed *chaîne opératoire* of sugar pots from the three areas considered, some common patterns were identified: the conical shape with a hole at the bottom; the use of calcareous raw material, and a raw material used for other wares (but not for cooking pots); a whitened surface; the use of the wheel-throwing method, at least for some part of the manufacture; the manufacture of the cones by sequential section. Some of these choices are directly related to the function of these pots. Their conical shape provides an easier release of the crystallised sugar loaf, and the hole at the bottom enables the discharge of the excess liquid. The use of the wheel-throwing method allows faster production and vessels having more standardised dimensions. A calcareous raw material produces a microporosity that may have favoured heat dissipation [22]. Conversely, some features were not related to the function of sugar pots; rather, they were part of the *habitus* of the potters; for example, the use of whitening the vessel surface can be also encountered in other wares and for more or less calcareous pastes. Some of these choices are linked to the manufacturing context in which these vessels were made, and cannot be explained only with regard to their function. In addition to these commonalities, the examined sugar pots diverge in terms of the characteristics of the clays (more or less calcareous), the firing strategy (generally fired at high temperatures, but also at low temperatures), the means of obtaining the white surface (use of seawater, or perhaps a raw material naturally rich in soluble salt), and the choice of the dimensions and design of the vessels, which were the most visible aspects [16]. These differences have a geographical correlation, i.e., sugar pots in Palermo are distinct from those made at Himera–Buonfornello and from those of Partinico, but are also site related. In Palermo, we can observe a variety of choices in the making of sugar pots, which can only in part be explained by a chronological aspect, and may be related to the organisation of the production of these pots. Although this point needs further research to integrate our results with archival sources, we can suggest two different scenarios. In the case of Palermo, the ceramic manufacture area is separated from that of the sugar production: sugar pots from different workshops reached the *trapetum*, but there was no immediate feedback to the potters about their product (Figure 19). In the case of Himera–Buonfornello, ceramic production took place in the *trapetum*: although potters did not neglect the production of other wares, as the case of ware produced in Polizzi shows, their production was focused on the needs of the sugar industry, and they may have worked together with other craftspeople in the *trapetum*. Although a certain degree of variability is observed in the production of in Himera–Buonfornello, the design and manufacturing sequences are strikingly more homogeneous than those of Palermo. It may be suggested that the community of practice at Himera–Buonfornello was focused on sugar production, whereas that in Palermo was focused on ceramic production itself.

**6. Conclusions**

Due to the pioneering research of archaeologists and historians who have shed light on many aspects of sugar production in Sicily, this project could move forward and consider the archaeological evidence from a different point of view, aided with an analytical examination. Three main areas of production of sugar pots were defined: Palermo, Himera–Buonfornello,

and one, for the present time, labelled Partinico. Completing the information from archival sources that refer to sugar pots circulating from other places to Palermo, this study revealed that sugar pots also circulated from Palermo to the other two areas. Further research is needed to discern whether these sugar pots were travelling filled with sugar to be consumed, or empty, to be filled with sugar and then returned to Palermo. The underwater discovery, probably also made in Sicily, shows that sugar pots were also circulating by sea.

The ways in which sugar pots were designed and manufactured followed some common features, such as the conical shape, the forming methods, and the adoption of calcareous pastes. The specific function of these vessels clearly constrained some potters' choices. Conversely, some technological characteristics were specific to each area, such as the firing strategies, the vessel profile, and the means of obtaining a white surface. In terms of the analysis, vessels showed similar heat transfer properties when fired at high temperatures; however, the vessel of Himera–Buonfornello were more capable of resisting thermal shock and crack initiation, whereas Palermo's vessels were probably tougher. When considering the forces these were subjected to during use, sugar cones from Himera and Palermo would perform in the same way when the hot syrup was poured inside, but those of Himera would remain stronger during continuous handling and travelling. Sugar cones produced in Palermo, in contrast, would crack more easily but withstand longer use after use. These differences reinforce the hypothesis that there does not seem to be a "standard" for sugar pot design and technology; rather, a local re-interpretation of common and generic requirements can be observed [16]. These idiosyncrasies may have originates in the context of manufacture; that is, the choices made by potters in making other wares and in the production organisation. The community of practice of potters in Palermo working on a wide range of ceramic products may have been very different from that of Himera–Buonfornello, where sugar pot manufacture was closely related to sugar production. The variety observed in Palermo sugar pots compared to those of Himera–Buonfornello may be explained by the different levels of connection of the ceramic manufacture to the sugar production. For Partinico, we cannot generate any hypotheses on this aspect at present due to the scarcity of the materials found.

As in the example of the can opener, would the reconstructed design and technological differences have a consequence for the actual performance of the vessels? Would their utilisation differ? Finite element analyses have been used in other cases to solve these questions [24,25], and will be further applied to the sugar pot case study.

Robertson argues that the terms globalisation and glocalisation should not be seen as opposing. He also argues that these terms involve the 'simultaneity and the interpenetration of what are conventionally called the global and the local or . . . the universal and the particular' [1] (p. 30). We do not argue in this paper whether sugar production and consumption in the Mediterranean can be considered a 'globalised' phenomenon; however, this concept also fits well in explaining the response of local potters to the particular demands for specific vessels from the sugar industry: potters thought globally but acted locally.

**Supplementary Materials:** The following supporting information can be downloaded at: https://www.mdpi.com/article/10.3390/min12040423/s1, Supplementary Material 1: Table S1 List of individuals examined and the analyses performed; Table S2 Correlation of vessel shape by context with the petrographic fabric as defined in Supplementary Materials 2; Table S3 Correlation of vessel shape by context with the chemical groups. Supplementary Material 2: Sample preparation and instrumental conditions and the petrographic descriptions.

**Author Contributions:** Conceptualization, R.M.; methodology, R.M., M.M.i.F., A.H., V.K. and J.B.i.G.; validation, R.M., M.M.i.F. and A.H.; formal analysis, R.M., M.M.i.F., A.H., V.K. and J.B.i.G.; investigation, R.M.; resources, R.M.; data curation, V.K. and J.B.i.G.; writing—original draft preparation, R.M., M.M.i.F. and A.H.; writing—review and editing, R.M., M.M.i.F., A.H., V.K. and J.B.i.G.; visualization, R.M., M.M.i.F. and A.H.; supervision, R.M., J.B.i.G. and V.K.; funding acquisition, R.M., J.B.i.G. and V.K. All authors have read and agreed to the published version of the manuscript.

**Funding:** This research is part of the project "Sugar Pot manufacture in western Europe in the medieval and post-medieval period (11–16th centuries AD)", funded under the Horizon 2020 Marie Skłodowska-Curie actions (grant agreement: 797242).

**Data Availability Statement:** The raw data presented in this study are openly available in the CORA, Research Data Repository.

**Acknowledgments:** For their support, their feedback and to provide access to the materials, we would like to thank L. Arcifa of Universita di Catania; F. D'Angelo, medieval archaeologist; E. Pezzini, F. Spatafora, C. Greco of Museo A. Salinas di Palermo; S. Vassallo, G. Battaglia e M.R. Cucco of Soprintendenza ai BBCCAA di Palermo; M.R. Panzica of Museo P. Marconi di Himera; G. Falsone of Universitá di Palermo; A. Molinari of Università La Sapienza di Roma; M. Gasparo and G. Montana of Universitá di Palermo for helping in identifying and collecting the clays from Partinico. The CCTIUB of Universitat de Barcelona for performing the analyses with special mention to M. Romero. G. Vekinis, head of the Advanced Ceramics and Composites Laboratory of the NCSR Demokritos for facilitating the use of the INSTRON 5982. The Labstone laboratory for preparing the thin sections. Judith, Julia, Marta and Cristina of the ARQUB and Maria and Dimitra of the NCSR Demokritos team for their scientific and, more importantly, human support. V. Testolini, V. Anicetiand S. Valenzuela for providing feedback and support since the first stages of the project.

**Conflicts of Interest:** The authors declare no conflict of interest. The funders had no role in the design of the study; in the collection, analyses, or interpretation of data; in the writing of the manuscript, or in the decision to publish the results.

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
