# Peer review of "Think Globally, Act Locally: Global Requirements and Local Transformation in Sugar Pots Manufacture in Sicily in the Medieval and Post-Medieval Periods"

_minerals, doi:10.3390/min12040423_

Round 1

Reviewer 1 Report

This is an impressive paper:   for the first time,  sugar vessels of the Medieval and post-Medieval periods – in this case from Sicily - have been treated in a holistic manner with particular emphasis on their material characteristics which range from mineralogical to thermo-mechanical properties and how the vessels were formed.  This welcome approach allows the authors to investigate questions of origin, technology and resilience alongside the vessels’ cultural context and in particular to consider the critical question posed by the authors whether “the potters thought globally but acted locally”.  The paper succeeds admirably in answering this question, at least for the material examined from NW Sicily.

The paper demonstrates a good awareness of current archaeological issues in material culture studies which are taken further in a more technological context in 3.1.  Following the full Introduction is the informative summary in Section 2  of the vessels’ archaeological context in Sicily with appropriate illustrations.  The results of analyses from the battery of techniques applied to the sugar pots (and modern clays) are fully presented and of high quality.  As would be expected from the authors who collectively have very considerable experience of archaeological ceramics, interpretations have been made confidently and realistically.  The long discussion is usefully divided into sections.  Discussion in 5.3 of the Palermo industry might benefit from a plan indicating the locations mentioned in the text beyond what is shown in Fig. 2c. 

In recommending the paper be accepted for publication, I have two main comments to which I ask the authors to respond: (1) the paper is long, probably over-long.  I think the text could be tightened up, and some of the detail in the Results could be transferred to the Supplementary files. (2) I suggest the premise that sugar pots were special in that their properties were carefully related to their intended function has been overplayed:  yes, these pots, especially the cones, had to have strength and certain thermal properties but the temperatures concerned – up to 200C but not higher – are well below those that are relevant to metallurgical ceramics and even cooking pots whose fabrics we know required the potter's intervention.  In that sense, sugar pots differed from the standard ceramic repertoire of the time more in terms of shape – the cone was a new shape – than in terms of a specific clay recipe or firing regime.  In this connection, although I am uncertain whether there is published comparative EFT data for the common medieval/post-medieval pottery in Sicily, I think it unlikely the sugar pots were fired at consistently higher temperatures;  this is partly because at some/many workshops the sugar pots were likely fired together with the ‘common' pottery.   In any case, this point does not detract from the axiom “the potters thought globally but acted locally”.

 The English is generally good, but I indicate below where I suggest corrections and rephrasing.

Specific points

Reference could usefully be made at some point to a comparable study: typological, petrographic and chemical analysis of sugar pots in the Akko Plain, Israel, showing changes in shape and composition from the 11th century to Ottoman (Shapiro, A., Stern, E.J., Getzov, N., Waksman, S.Y.  2020. ‘Ceramic evidence for sugar production in the ‘Akko plain: typology and provenance studies’. In S.Y. Waksman (Ed.) Multidisciplinary approaches to food and foodways in the medieval Eastern Mediterranean. Archéologie(s) 4, MOM Ä–ditions, Lyon, 163-189).

Some references in the bibliography have page numbers added, but this is not necessary in the case of books, eg. Mintz 1986.

140  sugar cones are often visualised as sitting on top of molasses jars, but it is quite likely that because of the cone’s weight and size  the cone would have toppled over;  instead, as shown at Tell abu Sarbut in Jordan,  the cone sat in a hole in the bench and the molasses jar placed separately under the cone.

219   Explain noria and cantaro vessels

396-412     This reviewer found this section  obscure with few of the equations’ parameters  realistically explained.

438   more or most variability?

441  How was the data transformed?

Fig. 6   how does the classification in the dendrogram compare when the Palermo data are treated on their own by cluster or principal components analysis?

            It is unclear where the samples belonging to groups PAL01 and PAL02 in Table 3 appear in the dendrogram in Fig. 6.    In the text these groups have a CG prefix which is confusing. 

608   A nice plot but surely we know most Mediterranean ceramics fall in the quartz-anorthite-wollastonite triangle.

698-700    I don’t understand the sentence ‘Neither the peak of calcite……

717-719  Reword the obscure sentence  ‘For none of them…..

732   due to the expenses of CaO    meaning?

916-918    This study confirms….  I don’t understand this sentence

923-925   I agree with this sentence. 

Grammatical and other points

5    ….i Garrigós, Jaume

31    … a difficult matter.  This is a difficulty that is not only practical….

33 in the past somehow, while the global….

35   … in design, although they probably will be…

82… revolution, as has been called the set of…., has been critically reviewed lately….

86-87     I don’t understand the sentence  ‘Some authors even suggest….   (Mintz 1985 or 1986?

88        … are now starting to reveal…

93   types of machinery (i.e. mill, a press,….

101   … the first activity in the Mediterranean            what first activity?

137    … sugar pots would require resistance to the thermal…..

197   Another 7 cones were found…..

211     …. Phases, one between the 11th and 13th centuries…..be put; another from the end of….                 Delete AD

218     87 sugar pots……… sugar cones), as well as 5 vessels….

230   and elsewhere    change geologic to geological

372      crashing or crushing?

492-494  …  photomicrograph in XP….

514   but they differ anyway from….

559   interferences caused by….

573  in the vessel’s body (so-called….

579    show no to poor preferred…..

606   The triangular ceramic phase…..

616   Whitney and Evans…

623  the three illite-muscovite peaks, which are usually present…..

698   … might not explain the firing temperature changes.

708     developed for F5, a fabric in which….

736…. This suggests that the significant…

Table 4  unfortunate splitting of words: vitrification and margins

794    As expected, the fired geological deposit…

839   … versus firing temperature and…

870   … sugar production were mainly….

872   …Partinico,  the archival record refers to….

921    … that potters always added….

951    a comparably though behaviour…      meaning?

P64      distinguished or distinguishable final products?

980-981    … the manufacture of the city has been tackled….      manufacturing capacity or capability of the city? 

1001    He mentions that the clay sources…..

1028     The use of the wheel-throwing method….

1030-1033      On the other hand…..                    reword this long and confusing sentence

1073-1077     Despite being the results of…..           reword this long and confusing sentence

Author Response

Dear reviewer,

we would like to thank you for your kind comment and your feedback which helped us improve the quality of the paper. We have made the changes requested and discussed the points your raised in the following text.  

Best Regards

the authors

This is an impressive paper:   for the first time, sugar vessels of the Medieval and post-Medieval periods – in this case from Sicily - have been treated in a holistic manner with particular emphasis on their material characteristics which range from mineralogical to thermo-mechanical properties and how the vessels were formed.  This welcome approach allows the authors to investigate questions of origin, technology and resilience alongside the vessels’ cultural context and in particular to consider the critical question posed by the authors whether “the potters thought globally but acted locally”.  The paper succeeds admirably in answering this question, at least for the material examined from NW Sicily.

The paper demonstrates a good awareness of current archaeological issues in material culture studies which are taken further in a more technological context in 3.1.  Following the full Introduction is the informative summary in Section 2  of the vessels’ archaeological context in Sicily with appropriate illustrations.  The results of analyses from the battery of techniques applied to the sugar pots (and modern clays) are fully presented and of high quality.  As would be expected from the authors who collectively have very considerable experience of archaeological ceramics, interpretations have been made confidently and realistically.  The long discussion is usefully divided into sections.  Discussion in 5.3 of the Palermo industry might benefit from a plan indicating the locations mentioned in the text beyond what is shown in Fig. 2c. 

This is a good point and something we thought about when writing the paper. We made a map based on the archival information in Ouerfelli 2008 and D’Angelo 2016, which however we can’t verify as-is out of our expertise and the scope of the paper. Future collaboration with these authors will be set to see whether we can further support the hypothesis of the different collations of sugar and ceramic production in Palermo.

In recommending the paper be accepted for publication, I have two main comments to which I ask the authors to respond: (1) the paper is long, probably over-long.  I think the text could be tightened up, and some of the detail in the Results could be transferred to the Supplementary files.

We know that the paper is a bit long, as it involves many techniques. We have tried to put as much as we can in the Supplementary materials and the raw data dataset, and we further moved the part on sample preparation and instrumental conditions in Supplementary material 2, leaving only a small paragraph in the main text.

 (2) I suggest the premise that sugar pots were special in that their properties were carefully related to their intended function has been overplayed:  yes, these pots, especially the cones, had to have strength and certain thermal properties but the temperatures concerned – up to 200C but not higher – are well below those that are relevant to metallurgical ceramics and even cooking pots whose fabrics we know required the potter's intervention.  In that sense, sugar pots differed from the standard ceramic repertoire of the time more in terms of shape – the cone was a new shape – than in terms of a specific clay recipe or firing regime.  In this connection, although I am uncertain whether there is published comparative EFT data for the common medieval/post-medieval pottery in Sicily, I think it unlikely the sugar pots were fired at consistently higher temperatures; this is partly because at some/many workshops the sugar pots were likely fired together with the ‘common' pottery.   In any case, this point does not detract from the axiom “the potters thought globally but acted locally”.

Thanks for this rising this point. The fact that the sugar cones were made differently due to their function is considered a starting hypothesis to test or go against with (we hoped to make it clear in the theoretical approach section and in line 148-150). In the discussion , we tackle this by stating that some of the features are functional but others depend on ceramic production context. So, the initial hypothesis is not fully supported. In terms of affordances, it is true that the temperature to which ceramic cones were exposed is lower than that of cooking pots and crucibles (which show different technological choices indeed). However, sugar cones had to withstand thermal shock when the hot liquid was filled in, and cooling and crystallisation had to be controlled by suitable heat transfer. It cannot be compared to cooking pots and crucibles, but it still needs to be explored before rejecting the hypothesis that they were made with an intended function in mind.

Regarding the EFT of other contemporary wares, we agree that sugar pots were fired probably with other wares. Noria vessels and cantaros show no difference compared to sugar pots, for example. However, there is a difference between Palermo, Partinico and Himera’s production, which might tell us something about organisation production organisation. We tried to compare our results with the available analyses, but we agree that we miss the whole picture for the same phases.

 The English is generally good, but I indicate below where I suggest corrections and rephrasing.

Specific points

Reference could usefully be made at some point to a comparable study: typological, petrographic and chemical analysis of sugar pots in the Akko Plain, Israel, showing changes in shape and composition from the 11th century to Ottoman (Shapiro, A., Stern, E.J., Getzov, N., Waksman, S.Y.  2020. ‘Ceramic evidence for sugar production in the ‘Akko plain: typology and provenance studies’. In S.Y. Waksman (Ed.) Multidisciplinary approaches to food and foodways in the medieval Eastern Mediterranean. Archéologie(s) 4, MOM Ä–ditions, Lyon, 163-189).

Thanks for the suggestion, we added the reference on in l. 91 and l. 128.

Some references in the bibliography have page numbers added, but this is not necessary in the case of books, eg. Mintz 1986.

We have revised it. We want to keep the pages reference for books as the reader can check where the information was taken. When a book is cited with no page specification, we are referring to a general concept tackled in the book.

140  sugar cones are often visualised as sitting on top of molasses jars, but it is quite likely that because of the cone’s weight and size  the cone would have toppled over;  instead, as shown at Tell abu Sarbut in Jordan,  the cone sat in a hole in the bench and the molasses jar placed separately under the cone.

Thanks for the comment. Yes, this is also how it is depicted in the illustration of Jan van der Straet and Motril. We think that the longer cone types were probably sitting in a hole in the bench. Not sure if that would be the same for the shorter and wider mouth types who could stand directly on the molasses jars, as the Steri from Palermo. Anyways, a sentence was added to encompass the two ways.  

219   Explain noria and cantaro vessels

Added a note with references, thanks.

396-412     This reviewer found this section obscure with few of the equations’ parameters realistically explained.

Some changes in the text has been done. More explanation is available in the literature cited

438   more or most variability?

Most of the variability; Changed

441  How was the data transformed?

Thanks for pointing it out. We added how the info

Fig. 6   how does the classification in the dendrogram compare when the Palermo data are treated on their own by cluster or principal components analysis?

Thanks for the comment. Data from the sites in Palermo (+Partinico as these are just a few individuals) were indeed treated independently before treating all the data together.  The Euclidean distance between the individuals doesn’t change: UND001 and STE004 continue to be set aside (due to contamination), the group from Partinico, and two groups including the same individuals found in PAL01 and PAL02. The same can be said for Himera data when treated independently. We added a sentence at l. 439 to be sure that the reader knows that.

            It is unclear where the samples belonging to groups PAL01 and PAL02 in Table 3 appear in the dendrogram in Fig. 6.    In the text these groups have a CG prefix which is confusing.

Thanks for pointing it out; the labels in the table have been revised. We are conscious that adding the prefix CG (chemical group) or P (petrography) makes it no easier for the reader. Still, these are the prefixes of the groups in the database of the ARQUB team where we needed to distinguish chemical and petro groups and which shortly will be online. We hope that what seems now very chaotic will facilitate the reader in the future when looking at the paper and the database!

608   A nice plot but surely we know most Mediterranean ceramics fall in the quartz-anorthite-wollastonite triangle.

It is very common indeed if we consider calcareous ceramics (even though we have cases where they fall in the Gh, An, Wo triangle). Still, the diagram gives us a starting hypothesis on which minerals will develop during firing before tackling the diffractograms, and it shows how the samples distribute.

698-700    I don’t understand the sentence ‘Neither the peak of calcite……

Changed for: “. As well as, the presence of the peak of calcite, which is prominent in both fabrics, cannot be used for the estimation of a low EFT: PE indicates that almost all these individuals show the presence of calcite recrystallized after firing.”

717-719  Reword the obscure sentence  ‘For none of them…..

Changed for: “None of the individuals from these fabrics shows the presence of the illite-muscovite peak; the peak of gehlenite is absent in F6 and F8, and the peak of calcite is completely decomposed in F8.”

732   due to the expenses of CaO    meaning?

Changed for: “causing a decrease of CaO”

916-918    This study confirms….  I don’t understand this sentence

Yes, it was a bit clumsy. Changed for: “In the case of the sugar pots from Himera – Buonfornello, the addition of seawater could have purposed the production of a white surface rather than to mitigate the presence of calcium-rich inclusions, which are not very common in the paste.”

923-925   I agree with this sentence. 

J

Grammatical and other points

All assessed, thanks

5    ….i Garrigós, Jaume

31    … a difficult matter.  This is a difficulty that is not only practical….

33 in the past somehow, while the global….

35   … in design, although they probably will be…

82… revolution, as has been called the set of…., has been critically reviewed lately….

86-87     I don’t understand the sentence  ‘Some authors even suggest….   (Mintz 1985 or 1986?

We removed the sentence as it was adding more “meat on the grill” in an already intricate discussion!

88        … are now starting to reveal…

93   types of machinery (i.e. mill, a press,….

101   … the first activity in the Mediterranean            what first activity?

137    … sugar pots would require resistance to the thermal…..

197   Another 7 cones were found…..

211     …. Phases, one between the 11th and 13th centuries…..be put; another from the end of….                 Delete AD

218     87 sugar pots……… sugar cones), as well as 5 vessels….

230   and elsewhere    change geologic to geological

372      crashing or crushing?

492-494  …  photomicrograph in XP….

514   but they differ anyway from….

559   interferences caused by….

573  in the vessel’s body (so-called….

579    show no to poor preferred…..

606   The triangular ceramic phase…..

616   Whitney and Evans…

623  the three illite-muscovite peaks, which are usually present…..

698   … might not explain the firing temperature changes.

708     developed for F5, a fabric in which….

736…. This suggests that the significant…

Table 4  unfortunate splitting of words: vitrification and margins

794    As expected, the fired geological deposit…

839   … versus firing temperature and…

870   … sugar production were mainly….

872   …Partinico,  the archival record refers to….

921    … that potters always added….

951    a comparably though behaviour…      meaning?

P64      distinguished or distinguishable final products?

980-981    … the manufacture of the city has been tackled….      manufacturing capacity or capability of the city? No, just the spatial organization

1001    He mentions that the clay sources…..

1028     The use of the wheel-throwing method….

1030-1033      On the other hand…..                    reword this long and confusing sentence

1073-1077     Despite being the results of…..           reword this long and confusing sentence

Reviewer 2 Report

This is an excellent, original piece of research that I warmly recommend for publication in Minerals. Medieval sugar pots manufactured in Sicily are discussed in the paper and the research team has done an excellent job at contextualising it and characterising it by multiple analytical methods, including both analytical and archaeological implications competently. The work is based on solid analytical foundations which are wisely used to understand petrographic, chemical and mineralogical composition patterns of the pots paste. Besides, the combination of different techniques allows the study of provenance, raw material manipulation, forming, firing regimes and surface treatments. Furthermore, presented results and discussion permits the identification of three main areas of production of sugar pots in Palermo, Himera-Buonfornello and Partinico. Also, this research exceeds the particular/regional focus of the Sicilian archaeological context and target broader international archaeological problems like the implications of the ´globalised´ sugar trade in the Medieval Mediterranean. In my opinion, it represents an interesting and useful approach to those topics. Besides, the general structure of the paper is clear, research objectives are worth and the summarized conclusions give to the reader (specialised or not) an easy way to reach the author’s approach.

Author Response

Dear reviewer,

we would like to thank you for your kind comment and your feedback which helped us improve the quality of the paper.

Best Regards

the authors

Reviewer 3 Report

missing (f) at 566 line

Author Response

Dear reviewer,

we would like to thank you for your comment and your feedback which helped us improve the quality of the paper. We have made the changes requested.  

Best Regards

the authors